# Profiling IgG and IgA antibody responses during vaccination and infection in a high-risk gonorrhoea population

Lenka Stejskal [1,8], Angela Thistlethwaite[1,8], Fidel Ramirez-Bencomo [1,8], Smruti Rashmi [1], Odile Harrison [2], Ian M. Feavers [3], Martin C. J. Maiden [3], Ann Jerse [4], Grace Barnes [5], Oscar Chirro[6], James Chemweno[6], Eunice Nduati[6], Ana Cehovin[5], Christoph Tang [5] ✉, Eduard J. Sanders [7] ✉ & Jeremy P. Derrick [1] ✉

Development of a vaccine against gonorrhoea is a global priority, driven by the rise in antibiotic resistance. Although *Neisseria gonorrhoeae* (Ng) infection does not induce substantial protective immunity, highly exposed individuals may develop immunity against re-infection with the same strain. Retrospective epidemiological studies have shown that vaccines containing *Neisseria meningitidis* (Nm) outer membrane vesicles (OMVs) provide a degree of cross-protection against Ng infection. We conducted a clinical trial (NCT04297436) of 4CMenB (Bexsero, GSK), a licensed Nm vaccine containing OMVs and recombinant antigens, comprising a single arm, open label study of two doses with 50 adults in coastal Kenya who have high exposure to Ng. Data from a Ng antigen microarray established that serum IgG and IgA reactivities against the gonococcal homologs of the recombinant antigens in the vaccine peaked at 10 but had declined by 24 weeks. For most reactive OMV-derived antigens, the reverse was the case. A cohort of similar individuals with laboratory-confirmed gonococcal infection were compared before, during, and after infection: their reactivities were weaker and differed from the vaccinated cohort. We conclude that the cross-protection of the 4CMenB vaccine against gonorrhoea could be explained by cross-reaction against a diverse selection of antigens derived from the OMV component.

The Gram-negative diplococcus *Neisseria gonorrhoeae* (Ng) causes the sexually transmitted infection gonorrhoea following infection of the epithelial cells of the genitourinary tract, but it can also colonise the ocular, nasopharyngeal, and anal mucosa. Sexually transmitted infections (STIs) constitute a major global health problem. The World Health Organisation estimated that approximately 82.4 million people were newly infected with Ng in 2020, making it the second most prevalent STI worldwide after *Chlamydia trachomatis* (CT)[1]. In addition, levels of resistance in Ng to frontline antibiotics is high and increasing[2]. The Sub-Saharan African region is reported to have the highest

[1]School of Biological Sciences, Manchester Academic Health Science Centre, The University of Manchester, Manchester M13 9PL, UK. [2]Nuffield Department of Population Health, University of Oxford, Oxford OX3 7LF, UK. [3]Department of Biology, 11a Mansfield Road, University of Oxford, Oxford OX1 3SZ, UK. [4]Department of Microbiology and Immunology, Uniformed Services University, 4301 Jones Bridge Road, Bethesda, MD 20814, USA. [5]Sir William Dunn School of Pathology, University of Oxford, South Parks Road, Oxford OX1 3RE, UK. [6]KEMRI-Wellcome Trust Research Programme, Kilifi, Kenya. [7]The Aurum Institute, Johannesburg, South Africa. [8]These authors contributed equally: Lenka Stejskal, Angela Thistlethwaite, Fidel Ramirez-Bencomo. ✉e-mail: christoph.tang@path.ox.ac.uk; ESanders@auruminstitute.org; jeremy.derrick@manchester.ac.uk

reported rates of STIs[3]. Gonorrhoea impacts the reproductive health of women and the well-being of newborns, particularly in Africa, and is a strong co-factor in HIV transmission[4]. The development of an effective vaccine against gonorrhoea has been identified as the most effective way of responding to these challenges[5]. However, infection with Ng does not usually induce protective immunity and clinical trials of gonococcal vaccine candidates have, until recently, not been encouraging[6]. For example, a purified pilus-based vaccine, although safe, failed to protect against gonococcal urethritis[7].

More encouraging data has been published recently which suggests that the development of an effective vaccine against gonorrhoea is feasible. A retrospective case-control study found that a vaccine (MeNZB®) derived from the related bacterium *Neisseria meningitidis* (Nm) was linked to a reduction in gonorrhoea diagnoses; however, the estimated efficacy was modest, at 31%[8]. The vaccine contained an outer membrane vesicle (OMV) preparation derived from a serogroup B Nm strain and was originally introduced to control an outbreak of meningococcal infection in New Zealand. This observation implies that antigens within the Nm OMV vaccine-elicited antibodies that cross-reacted with gonococcal antigens. Further studies have sought to verify and extend this observation using another vaccine with established efficacy against meningococcal infection, 4CMenB (Bexsero)[9]. 4CMenB consists of an OMV preparation from the same Nm strain as MeNZB®, combined with five recombinant antigens, of which three are responsible for enhancing protection against Nm infection[10]. There is now evidence that 4CMenB provides cross-protection against gonorrhoea: for example, Wang et al. showed that the vaccine provided moderate protection against gonococcal infection in adolescents and young adults up to three years post-immunisation[11]. In a murine infection model, immunisation with 4CMenB has been shown to accelerate clearance and reduce the bacterial burden[12]. Antibodies from vaccinated animals identified several antigens, including integral outer membrane proteins such as PilQ, BamA, MtrE, Opa (opacity) proteins and the porin PorB, a major protein in the Ng outer membrane. Other studies have used proteomic[13,14] and bioinformatic[15] approaches to identify antigens that could be responsible for antibody cross-reaction. Semchenko et al. examined the cross-reactivity of antibodies using sera from rabbits or humans vaccinated with 4CMenB, with proteins derived from Ng OMVs[16]. IgG antibodies in rabbit sera recognised the gonococcal homologues of three of the five recombinant proteins added to the 4CMenB formulation: NHBA, GNA2091 and GNA1030. GNA2091 and GNA1030 were noted as having a high degree of identity between meningococcal and gonococcal homologues (over 90%), whereas NHBA (a heparin-binding protein) and Factor H binding protein have identity levels of 69 and 63% respectively[16]. The fifth recombinant antigen in 4CMenB, the adhesin NadA, is not present in Ng. In the same study, human sera from vaccinated individuals showed a reaction against several Ng OMV proteins post-vaccination.

These studies have identified a range of Ng OMV-derived proteins that could play a protective role as target antigens for antibodies induced through vaccination with 4CMenB. In an earlier study, we showed how a dedicated microarray of Nm antigens could be used to identify IgG responses against specific antigens induced by an OMV vaccine[17]. We extended this work by identification of correlations in antigen responses within the dataset[18]. This study also showed how metadata, in the form of serum bactericidal assay (SBA) measurements, could be used to identify which antigens were associated with bactericidal activity.

In this study, we adopt a similar approach: using antigen microarrays, combined with statistical and machine-learning methods, a comprehensive view of how antigen reactivity profiles are linked to vaccination with 4CMenB and infection in a highly exposed human population is obtained. Individually purified, recombinant gonococcal proteins are immobilised in microarrays onto a nitrocellulose-coated slide. The collective binding dataset for antibodies to specific antigens generates a complex reactivity profile for each serum sample. We then apply statistical, multidimensional data reduction and clustering methods to quantify relationships between samples or antigens. We selected a coastal Kenyan population at high risk of infection: there is evidence that individuals with repeated Ng exposure can develop a degree of protective immunity[19], although it should be noted that this has not been seen in all studies[20]. The results indicate that vaccination induces a different pattern of IgG and IgA responses compared with individuals who have had Ng infection, where antibody induction is more restricted and against different antigen subsets.

## Results

Between June 2021 and February 2022, a total of 62 participants were screened to receive 4CMenB, with 50 (37 men and 13 women) enroled and receiving their first vaccination. Subsequently, 47 participants (36 men and 11 women) received a second vaccination (three participants did not complete the second vaccination and were lost to follow-up). Of nine participants living with HIV (three men and six women), eight completed the study (three men and five women). Only one participant had a rectal Ng infection at study completion (GeneXpert positive only). Serum samples were obtained from 50 participants at t0 and 47 participants at weeks 10 (t10) and 24 (t24), including the eight participants living with HIV. A heatmap of the IgG and IgA responses to individual antigens shows that participants' sera had significant reactivity against some antigens, even before vaccination (t0, Fig. S1). Strong IgG responses to some antigens at t10 and t24 were immediately apparent, compared with baseline (t0). Similar patterns in IgA reactivities were not obvious and required further quantitative analysis.

The complete IgG reactivities can be treated as a multivariate dataset: in addition to IgG values for each antigen, the vaccine recipient identity and the vaccination stage (t0, t10 and t24) are independent categorical variables. Multivariate analysis of variance (MANOVA) conducts a regression analysis for the multiple dependent variables (IgG reactivities) for these categorical variables- of particular interest in this case is the effect of the vaccination stage on IgG reactivities against specific antigens. We found that ANOVA simultaneous component analysis (ASCA), developed initially as a tool for metabolomic research[21], was particularly useful for these antigen microarray datasets, at least in part because it allowed for a graphical representation of the data. ASCA is a combination of ANOVA statistical analysis with principal component analysis (PCA); PCA is a widely used dimensionality reduction method which captures the variation in complex datasets. Figure 1A shows the results of ASCA analysis applied to the IgG reactivities of all antigens: each point is an individual serum sample, plotted as the first two components which account for the variance in the dataset (in the case of Fig. 1A, this is a total of 99.6%). As is the case for PCA, each axis represents a component which captures the variation in the dataset, with the first component contributing the most, the second the next largest and so on. There is a clear shift in overall antigen responses from t0 to 2 weeks post-second vaccination (t10), separated horizontally by the first component which contributes most of the variance. This effect relaxes to some degree in the transition from t10 to t24, indicating that the magnitude of responses faded between these timepoints. For each vaccination stage variable (t0, t10 or t24), confidence ellipsoids were calculated- these represent confidence levels in variance for each timepoint[22]. Ellipsoids are plotted at three different effect level means (40, 68 and 95%): the distance between the ellipsoids gives an indication of the uncertainty in the separation of the categories in the statistical model. The antigens that contribute most to these separations are shown in the righthand panel in Fig. 1A: in particular, GNA2091 and NHBA have the highest loadings, accounting for the divergence in component 1. These two antigens, along with Factor H binding protein and GNA1030, comprise the

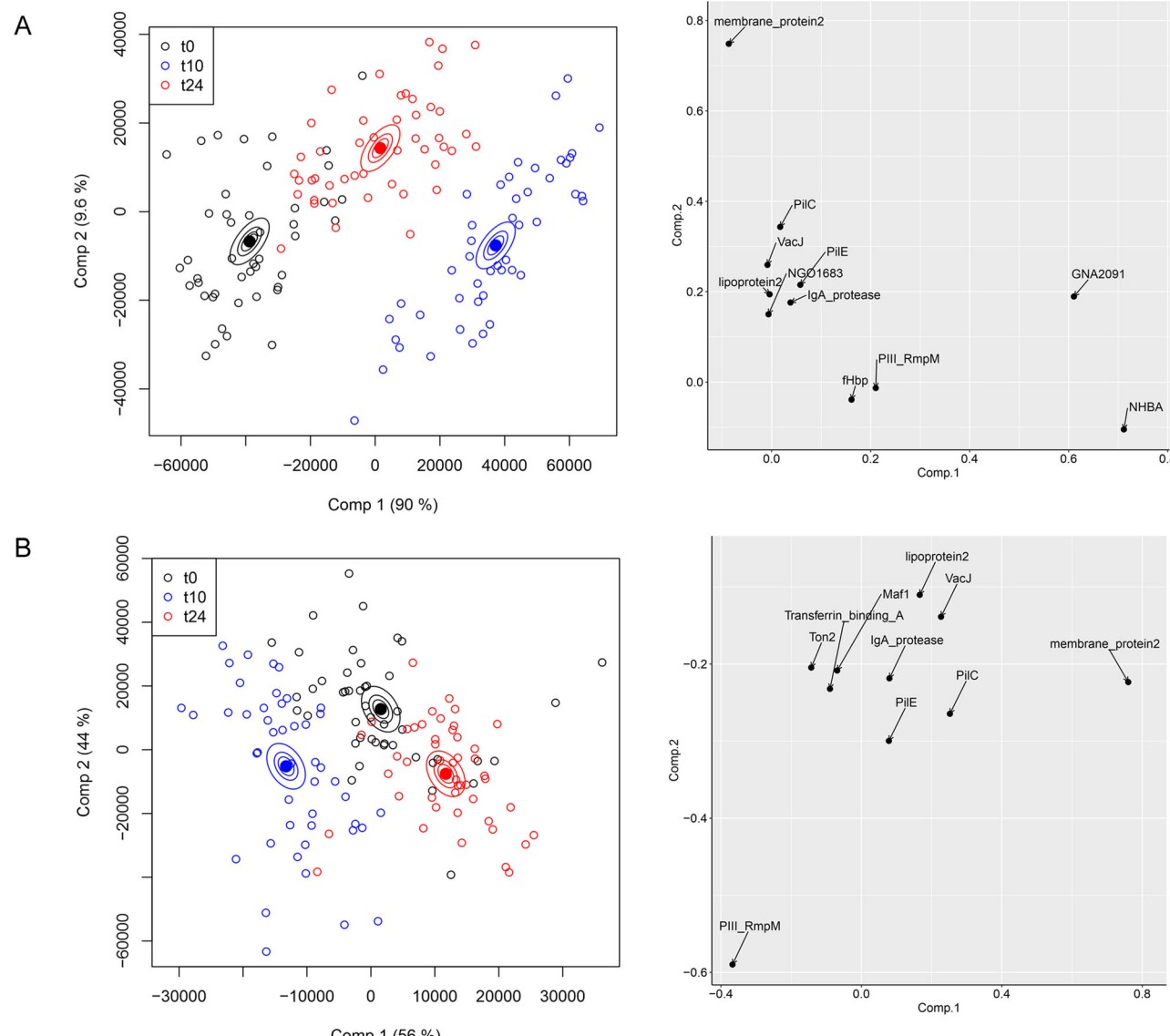

**Fig. 1 | ASCA separation of IgG antibody profiles by vaccine recipient.** In the left-hand panels, each point represents a profile derived from IgG reactivity against antigens across the array. Comp 1 and Comp 2 represent the first and second principal components in the ASCA analysis. Confidence ellipsoids are plotted that reflect the estimated variation of the effect level means at 40, 68 and 95%[22]. Open symbols represent individual serum samples; ellipsoid centres are plotted as filled symbols. Righthand panels show the antigens with the highest loadings for each component. **A** ASCA calculation with all antigens included. **B** As **A** but with NHBA, Factor H binding protein (fHbp), GNA2091 and GNA1030 removed.

gonococcal homologues of four of the five meningococcal recombinant antigens which are added to the OMV preparation in 4CMenB. It is apparent from the antigens separated in the second, vertical component in Fig. 1A that homologues of OMV-derived antigens (such as membrane protein 2 and PilC) contribute to the separation between t0, t10 and t24 samples. To remove the distorting effect of the recombinant antigens, the ASCA analysis was repeated with their contributions excluded (Fig. 1B). The separation between t0, t10 and t24 is still clear, although less than in Fig. 1A: this means that IgG reactivities against OMV-derived antigens differ between these time-points. Examination of the antigens with the highest loadings (right panel, Fig. 1B), reveals details of further contributing OMV-derived antigens, including the peptidoglycan-binding protein RmpM and the pilin protein PilE. The contributions of these antigens and their identities are analysed in further detail below.

The same approach was applied to serum IgA reactivities (Fig. 2). Ostensibly, the results are similar to those obtained for IgG, in that the first component separates the t0, t10 and t24 timepoints, with the t24 samples closer to t0. The first component, again, is dominated by NHBA and GNA2091 (Fig. 2A, right panel). However, the removal of the four recombinant antigens unexpectedly revealed that different OMV antigens are responsible for separation compared with IgG (Fig. 2B compared with Fig. 1B). This latter observation prompted us to examine whether the profiles of IgG and IgA responses to the panel of gonococcal antigens were different following the administration of 4CMenB. The results of applying t-distributed stochastic neighbour embedding (tSNE)[23] to examine the increase in IgG and IgA responses from pre-immunisation (t0) to the t10 and t24 samples are shown in Fig. 3A. tSNE is a similar method to PCA, in the sense that it is a multidimensional reduction method, although it works in a different way. In Fig. 3A, points close in space indicate that individuals across the vaccinated population responded similarly to a given antigen. It is apparent that, although overlapping, the profiles of IgG and IgA reactivities are distinct. Across the vaccinated cohort, the profile of

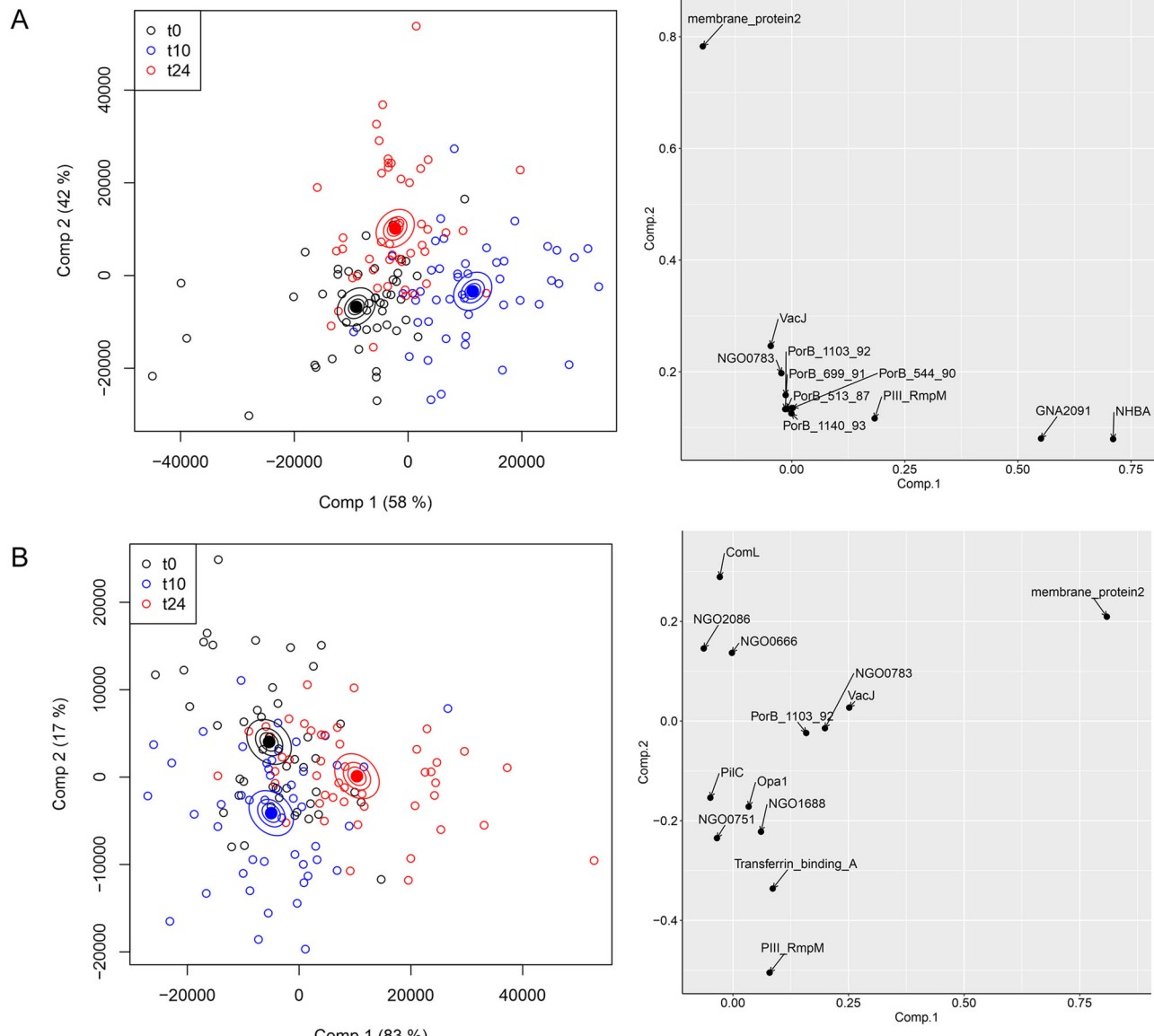

**Fig. 2 | ASCA separation of IgA antibody profiles by vaccine recipient.** In the left-hand panels, each point represents a profile derived from IgA reactivity against antigens across the array. Comp 1 and Comp 2 represent the first and second principal components in the ASCA analysis. Confidence ellipsoids are plotted that reflect the estimated variation of the effect level means at 40, 68 and 95%[22]. Open symbols represent individual serum samples; ellipsoid centres are plotted as filled symbols. Righthand panels show the antigens with the highest loadings for each component. **A** ASCA calculation with all antigens included. **B** As (**A**) but without the recombinant antigens, NHBA, fHbp, GNA2091 and GNA1030.

antigens against IgG was, therefore, appreciably different from IgA following vaccination. Possible explanations for this phenomenon could lie in B-cell proliferation arising in mucosal or systemic locations[24]. Specific antigens which were responsible for the greatest stimulation in IgG and IgA are identified from a detailed statistical analysis presented below.

The dataset was separated into IgG and IgA subsets and examined for responses to groups of antigens that were similar within those subsets. For example, our array contained ten different Opa proteins from Ng strain FA1090; this group of integral outer membrane proteins exhibited a high degree of sequence variability in their external loop regions. A BLAST search of the genome sequence of the strain used as the source of OMVs in 4CMenB (PubMLST ID 34542[25]) identified an *opa* allele with sequence identities to the ten FA1090 Opas which ranged from 60 to 68%. We noted that the IgG and IgA responses to the Opa proteins in our microarray tended to cluster in regions of the tSNE plots (Fig. 3B, C). To analyse

their relationships in more detail, the Opa points alone are plotted in Fig. S2, and a sequence alignment of the variants is given in Fig. S3. A phylogenetic tree of these sequences grouped OpaD/Opa9/Opa58, Opa1/Opa5/Opa8, Opa7/Opa3/Opa4, with Opa6 as an outlier. The most obvious alignment of these sequence groups with similarities in IgG or IgA reactivities was for the OpaD/Opa9/Opa58 grouping: these Opa proteins were consistently co-located in the tSNE plots for both IgG and IgA (Fig. S2). These plots are for the differences in responses for each antigen (eg t10-t0). It is also instructive to plot the raw data: as an example, plots of IgG and IgA reactivities for Opa9:OpaD and Opa58:OpaD are shown in Fig. S4. There is no evidence for stimulation of these antibodies by vaccination, which would be evident from a preponderance of red (t24) and blue (t10) data points at higher IgG or IgA values. However, the correlations in responses are clear, for both IgG and IgA. We conclude that there was a significant degree of antibody cross-reactivity between these Opa variants but that the B cells responsible for the

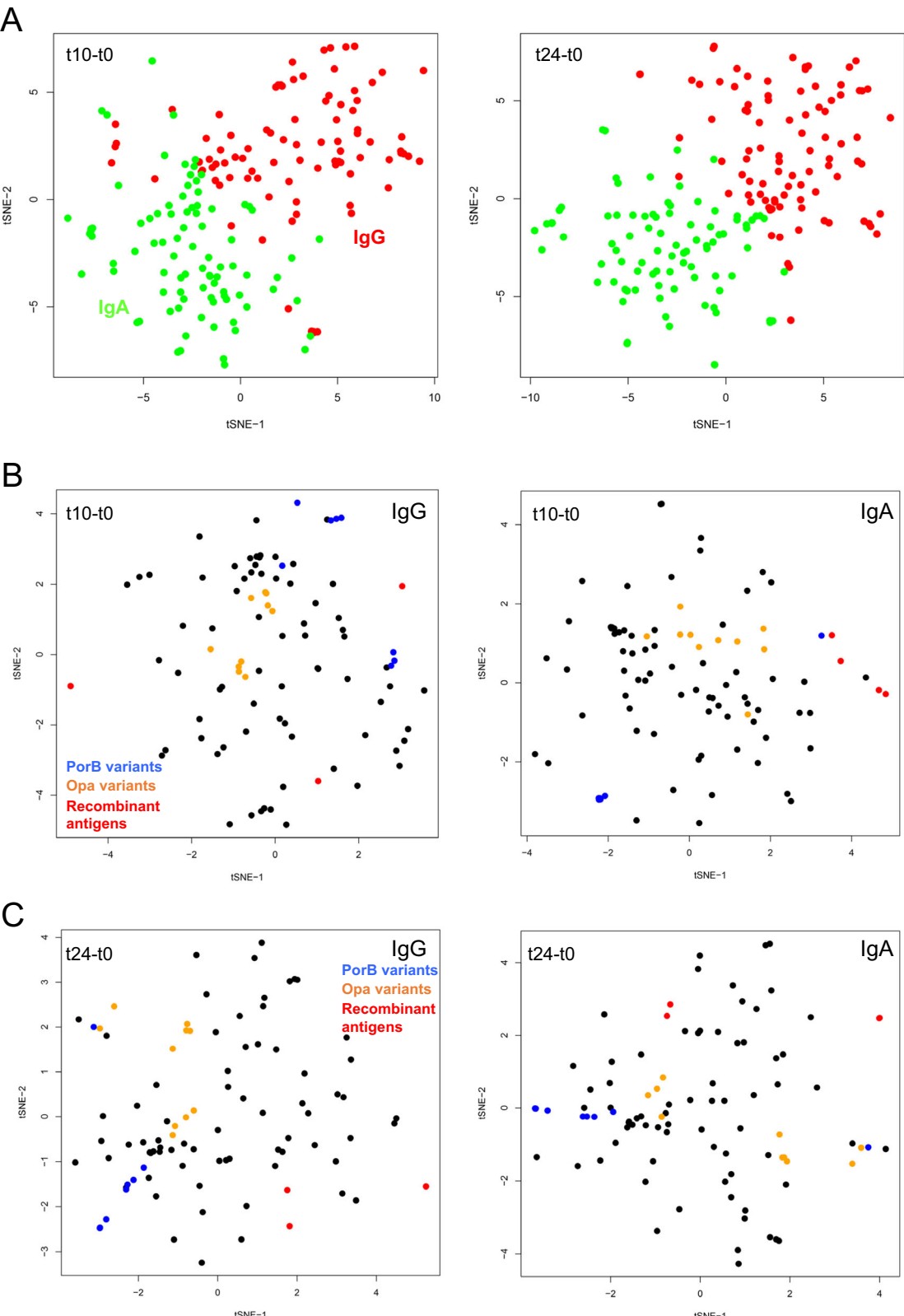

**Fig. 3 | tSNE separation of antibody profiles by antigen.** Each point represents an antigen profile derived from vaccinated individuals. **A** Comparison of IgG and IgA responses for stimulation of antibody reactivity against antigens between t0/t10 (left) and t0/t24 (right). IgG reactivities are coloured in red and IgA in green.

**B** Separation of IgG and IgA reactivities against array antigens for the t10-t0 dataset. The PorB variants are coloured in blue, the Opa variants in orange, the four 4CMenB recombinant antigens in red and all others in black. **C** as for (**B**), but for the t24-t0 dataset.

generation of these antibodies are not stimulated to proliferate by vaccination with 4CMenB.

The porin PorB is a major component of the outer membrane[26,27]; given its abundance, a total of eight PorB variants were included in the array, derived from diverse Ng strains. Despite their sequence diversity, we observed clustering of PorB variants within these plots (Fig. 3B, C). This observation was confirmed by the calculation of correlations between antigen responses. We also noted that both IgG and IgA responses to all PorB antigens were generally low. Percentage identities to the Nm PorB sequence in 4CMenB ranged from 70 to 78%, with most deviations mapping to the large external loop regions. Anti-PorB antibody responses induced by the meningococcal PorB against gonococcal variants were therefore low, but those antibodies exhibited a high degree of cross-reactivity between different gonococcal variants. Potentially, these responses might be targeted against more conserved loop regions. Separately, we noted that homologues of the four recombinant proteins in 4CMenB did not exhibit consistent evidence of clustering.

We conducted a detailed statistical analysis using the Wilcoxon paired rank test to identify the antibody responses against gonococcal antigens which were most stimulated by vaccination with 4CMenB. Four separate analyses were carried out, for IgG and IgA, making comparisons between t0 and t10, and t0 and t24 (Tables 1, 2). Overall, more antigens were identified from IgG responses compared with IgA, probably reflecting the weaker overall signals from IgA. We also noted that the t0:t24 comparison revealed more antigens than t0:t10, reflecting stronger and more sustained responses to 4CMenB.

The highest-ranked antigens in this analysis were homologues of two of the four recombinant proteins in the vaccine, NHBA and GNA2091: this was the case for both IgG and IgA, at both timepoints. The other two recombinant antigens, fHbp (Factor H binding protein) and GNA1030, featured lower down the table for IgG (Table 1) and, interestingly, were not among the antigens with the lowest $p$ values for IgA (Table 2). Visualising responses against individual antigens as violin plots (Fig. 4) illustrates that NHBA and GNA2091 gave much higher amplitudes than fHbp and GNA1030, particularly for IgG (Fig. 4, left boxes). It was also readily apparent that both IgG and IgA antibodies against NHBA and GNA2091 peak at t10 and decline substantially by t24. fHbp and GNA1030 exhibited a similar trend (Fig. 4), which contrasts with the prolonged responses to OMV-derived antigens.

IgG and IgA responses to antigens, other than homologues of the four recombinant antigens, are presumed to originate from cross-reaction against proteins in the OMV component of 4CMenB. Strikingly, many of these proteins were different between IgG (Table 1) and IgA (Table 2), in accordance with the tSNE analysis (Fig. 3A). We also found that, whatever $p$ value limit was selected for inclusion, more OMV-derived antigens met the criterion at t24 compared with t10. This observation contrasts with the behaviour of the recombinant antigens (Fig. 4), even though they are co-administered. The reactivity of a specific antigen can be influenced by the degree of sequence conservation between the meningococcal OMV antigen and its gonococcal counterpart on the microarray (Tables 1, 2). Most antigens, although not all, were at >90% amino acid identity, which is one-factor driving cross-reactivity. Several antigens in Tables 1, 2 did not have a close meningococcal homologue, raising the question of why IgG or IgA antibodies were generated against them. One explanation is that this cross-reactivity may be caused by localised regions of sequence and/or structural similarity which form conserved epitopes, possibly for functional reasons. Examples of antigens where this might be the case are membrane protein 1 (NEIS2704), which contains an OmpA-like domain, or NGO1847 (NEIS2724), which contains a Tetratricopeptide Repeat (TPR) repeat. TPR is a structural motif which is widespread in bacterial pathogens[28].

The antigens identified in Tables 1, 2 are diverse in function, size and structure. Some, such as the PorB-associated protein RmpM[29],

were highlighted in our previous analysis of individuals vaccinated with an OMV preparation without recombinant proteins[17,18]. Intrinsic membrane proteins with transmembrane β-barrel structures, like NGO1847, are featured in the list. Another notable feature was the inclusion of three proteins from the BAM complex- BamA, C and E- which promote outer membrane protein folding and insertion[30]. Several proteins involved in type IV pilus assembly are also listed: the secretin PilQ[31], the adhesin PilC[32], the inner membrane assembly platform protein PilN and, interestingly, the pilin protein PilE which constitutes the main component of the type IV pilus fibre. PilE is subject to a high degree of sequence variation through gene conversion[33] and has a relatively low level of identity (66%) between the pilin in the 4CMenB OMV and Ng strain FA1090, which was chosen as the source of antigens for the microarray. Nevertheless, there seems to be an induction of a significant population of cross-reactive IgG antibodies against PilE after vaccination with 4CMenB.

Violin plots for some of the selected OMV antigens in Tables 1, 2 allow direct comparison of IgG and IgA responses against the recombinant antigens (Figs. 5, 6). Some antigens induce responses which are comparable to, or even exceed, those of fHbp and GNA1030, notably PilC (NEIS0371) for IgG, and membrane protein 2 (NEIS1304) for IgG and IgA. Robust IgG responses were recorded for RmpM, the peptidoglycan-binding protein which interacts with PorB trimers[29], although PorB itself is not well represented, which is probably indicative of the comparatively high sequence diversity of PorB compared with RmpM.

We found little evidence to suggest that HIV infection affected the IgG or IgA immunoprofiles. The t10-t0 and t24-t0 datasets, which quantify the stimulation of IgG or IgA responses at the 10 and 24 weeks timepoints respectively, were subjected to PCA analysis in order to identify any systematic differences between individuals who were positive or negative for HIV infection (Figs. S5, S6). Each point is derived from the t10-t0 and t24-t0 immunoprofiles for each vaccine; points are coloured by either HIV status (positive/negative) or sex (male/female). Ellipses are superimposed at 95% probability. There is little evidence for separation between the groups, for IgG or IgA, at either timepoint, with the possible exception of HIV status at t10-t0, for IgG (Fig. S5, top left). However, even in this case, there is considerable overlap between the HIV-positive and negative groups. It should be noted that participants living with HIV were on antiretroviral therapy, and their viral load was suppressed. Similarly, there was no obvious effect of sex on IgG or IgA immunoprofiles.

Bactericidal responses were measured for all sera in the vaccinated cohort; comparison of overall % recovery for samples from the t0, t10 and t24 timepoints did not reveal any obvious differences with any of the three serum dilutions tested (Fig. S7). This observation was confirmed from an analysis using the Wilcoxon paired rank test (not shown). To examine whether any IgG or IgA responses correlated with bactericidal activity, we carried out an analysis using Spearman's rank correlation for all antigens at each of the three dilutions. Correlations were generally weak and did not produce consistent results in comparisons between measurements at the three dilutions (not shown). We conclude that there is no evidence for a relationship between IgG or IgA responses to individual antigens and bactericidal titres.

We used an ELISpot assay[34] to examine whether we could observe antigen-specific responses from T-cells, measuring IFN-γ production. Due to limitations in the supply of PBMCs, assays were limited to two pools of peptides, which were selected by an informatics screening process which predicted the strongest and most highly conserved T-cell epitopes in 22 different antigens (details in Materials and Methods). The highest-ranking peptides were derived from NHBA, PorB, PilQ, NspA and BamA and were allocated to two different pools (Table S1). For each predicted T-cell epitope peptide, a 'mock' peptide was synthesised, which contained the same residues, but in a different sequence. We did not, however, observe any statistically significant

**Table 1 | Summary statistical analysis of the most reactive gonococcal antigens against IgG following vaccination with 4CMenB**

| Label | Function | % identity[a] | NEIS[69] | 4CMenB Recombinant?[b] | Stage[d] | p value[c] |
|---|---|---|---|---|---|---|
| NHBA | Heparin-binding protein | 68[e] | NEIS2109 | x | t0:t10 | 6.47E-13 |
| GNA2091 | Hemolysin/lipoprotein | 96[e] | NEIS2071 | x | t0:t10 | 6.47E-13 |
| BamE | Outer membrane protein assembly | 93 | NEIS0196 | | t0:t10 | 6.37E-09 |
| lipoprotein 1 | Genome-derived Neisseria antigen 1162 | 97 | NEIS1063 | | t0:t10 | 2.54E-08 |
| Factor H binding protein | Binds to serum Factor H | 63[e] | NEIS0349 | x | t0:t10 | 2.57E-06 |
| PIII/RmpM | PorB stabilisation | 93 | NEIS1783 | | t0:t10 | 3.94E-06 |
| NGO1225 | Peptidyl-prolyl isomerase | 98 | NEIS1487 | | t0:t10 | 3.12E-04 |
| GNA1030 | Lipid/polyisoprenoid-binding; YceI-like | 93[e] | NEIS1183 | x | t0:t10 | 9.56E-04 |
| LolB | Lipoprotein localisation | 96 | NEIS0814 | | t0:t10 | 2.05E-03 |
| NGO0432 | Porin-like protein | 98 | NEIS0807S | | t0:t10 | 7.85E-03 |
| NHBA | Heparin-binding protein | 68[e] | NEIS2109 | x | t0:t24 | 6.54E-13 |
| GNA2091 | Hemolysin/lipoprotein | 96[e] | NEIS2071 | x | t0:t24 | 1.94E-12 |
| PIII/RmpM | PorB stabilisation | 93 | NEIS1783 | | t0:t24 | 5.19E-08 |
| Factor H binding protein | Binds to serum Factor H | 63[e] | NEIS0349 | x | t0:t24 | 3.13E-06 |
| IgA protease | IgA cleavage/immune evasion | 88 | NEIS0651 | | t0:t24 | 1.05E-05 |
| NGO1847 | Transmembrane protein with TPR repeat | _[f] | NEIS2724 | | t0:t24 | 1.32E-04 |
| GNA1030 | Lipid/polyisoprenoid-binding; YceI-like | 93[e] | NEIS1183 | x | t0:t24 | 2.73E-04 |
| PilC | Type IV pilus-related function | 73 | NEIS0371 | | t0:t24 | 4.40E-04 |
| membrane protein 2 | Membrane-associated lipoprotein | 97 | NEIS1304 | | t0:t24 | 4.89E-04 |
| Transferrin binding B | Iron uptake | 61 | NEIS1691 | | t0:t24 | 5.19E-04 |
| BamA | Outer membrane protein assembly | 88 | NEIS0173 | | t0:t24 | 6.40E-04 |
| lipoprotein 2 | BamC; outer membrane protein assembly | 97 | NEIS0906 | | t0:t24 | 6.78E-04 |
| NGO2086 | Surface-exposed, possible adhesin | _[f] | NEIS2733 | | t0:t24 | 7.10E-04 |
| NGO1040 | Porin-like protein | 92 | NEIS1404 | | t0:t24 | 7.10E-04 |
| PhospholipasePhospholipase | Hydrolysis of phosphatidylcholine | 99 | NEIS1687 | | t0:t24 | 1.03E-03 |
| LolB | Lipoprotein localisation | 96 | NEIS0814 | | t0:t24 | 1.74E-03 |
| LbpB | Iron uptake | 74 | NEIS1469 | | t0:t24 | 2.36E-03 |
| Maf1/MafA | Adhesin | 98 | NEIS2083 | | t0:t24 | 2.58E-03 |
| NGO1430 | TonB-coupled transporter | _[f] | NEIS2697 | | t0:t24 | 2.72E-03 |
| BamE | Outer membrane protein assembly | 93 | NEIS0196 | | t0:t24 | 2.82E-03 |
| PilE | Type IV pilin protein | 66 | NEIS0210 | | t0:t24 | 3.36E-03 |
| LptD | LPS assembly protein | 90 | NEIS0275 | | t0:t24 | 3.53E-03 |
| PilQ | Secretin; Type IV pilus assembly | 91 | NEIS0408 | | t0:t24 | 4.19E-03 |
| PorB_537_89 | Porin protein | 66 | NEIS2020 allele 537 | | t0:t24 | 7.23E-03 |
| TamB | Translocation/assembly | 97 | NEIS2113 | | t0:t24 | 8.64E-03 |

[a]% identity between FA1090 protein as immobilised on the array slide (with the His tag removed) and the equivalent protein in the NZ98/254 strain used in 4CMenB or closest equivalent in another strain.
[b]Denotes recombinant proteins in the 4CMenB formulation.
[c]Calculated using a one-sided Wilcoxon paired rank test with correction after Hommel[67], listed in increasing order of p value. Only antigens with p < 0.01 are listed.
[d]Paired t0 with t10 or t0 with t24.
[e]As reported by ref. 16.
[f]No close homologues in *Neisseria meningitidis*.

differences between stimulation by the experimental peptide pools in comparison with their mock controls (Fig. S8). This result could be due to inaccuracies in T-cell epitope prediction, or challenges in distinguishing antigen-specific responses against a background of natural biological variability.

To compare antibody responses in vaccinated with non-vaccinated individuals, we made use of a historical cohort of individuals with laboratory-confirmed gonococcal infection. These were collected from a similar Kenyan population to those involved in the 4CMenB study, comprising sex workers and MSM presenting at clinics over a 9-year period. This dataset comprised 160 samples: 52 were collected before the recorded infection, 63 during the infection

period (within 28 days) and 45 in the period after recovery (median period of 31 days). We refer to these groups as before, during, and after, respectively. The fact that multiple samples were available from the same individuals was valuable in helping to eliminate person-to-person variability in antibody reactivity profiles. Total IgG and IgA profiles were measured employing the same antigen microarray, and data analysed in a similar fashion to the vaccinated cohort (with before/during/after treated as timepoints t0/t10/t24, respectively).

Application of the Wilcoxon paired rank test did not produce statistical association of any individual antigen with before/during/after status with a p value below 0.05, after correction for multiple

**Table 2 | Summary statistical analysis of the most reactive gonococcal antigens against IgA following vaccination with 4CMenB**

| Label | Function | % identity[a] | NEIS | 4CMenB Recombinant?[b] | p value[c] | Stage[d] |
|---|---|---|---|---|---|---|
| NHBA | Heparin-binding protein | 68[e] | NEIS2109 | x | 9.69E-10 | t0:t10 |
| GNA2091 | Hemolysin/lipoprotein | 96[e] | NEIS2071 | x | 8.40E-09 | t0:t10 |
| PIII/RmpM | PorB stabilisation | 93 | NEIS1783 | | 7.07E-04 | t0:t10 |
| NGO0751 | Autochaperone domain-containing protein | -[e] | NEIS2658 | | 1.33E-03 | t0:t10 |
| GNA2091 | Hemolysin/lipoprotein | 96[e] | NEIS2071 | x | 2.62E-07 | t0:t24 |
| NHBA | Heparin-binding protein | 68[e] | NEIS2109 | x | 4.31E-07 | t0:t24 |
| NGO0690 | Lipoprotein | 86 | NEIS1164 | | 1.69E-05 | t0:t24 |
| PilN_domain | Type IV pilus assembly | 97 | NEISO411 | | 3.26E-05 | t0:t24 |
| membrane protein 2 | Membrane-associated lipoprotein | 97 | NEIS1304 | | 5.01E-05 | t0:t24 |
| NGO1063 | Membrane-associated lysozyme inhibitor (MliC) | 92 | NEIS1425 | | 7.92E-04 | t0:t24 |
| NGO0834 | Curli pilus assembly (CsgG-like) | 96 | NEIS1066 | | 1.40E-03 | t0:t24 |
| HpuB | Haemoglobin-haptoglobin utilisation protein B | 94 | NEIS1947 | | 1.48E-03 | t0:t24 |
| NGO0891 | Lipoprotein | 95 | NEIS2666 | | 1.75E-03 | t0:t24 |
| ComP | Pilin-like protein; Natural competence | 99 | NEISO899 | | 3.57E-03 | t0:t24 |
| membrane protein 1 | Lipoprotein; OmpA-like domain | -[e] | NEIS2704 | | 9.36E-03 | t0:t24 |

[a]% identity between FA1090 protein as immobilised on the array slide (with the His tag removed) and the equivalent protein in the NZ Nm strain used in 4CMenB or closest equivalent in another strain.
[b]Denotes recombinant proteins in the 4CMenB formulation.
[c]Calculated using a one-sided Wilcoxon paired rank test with correction after Hommel[67] listed in increasing order of p value. Only antigens with p < 0.01 are listed.
[d]Paired t0 with t10 or t0 with t24.
[e]No close homologues in *N. meningitidis*.

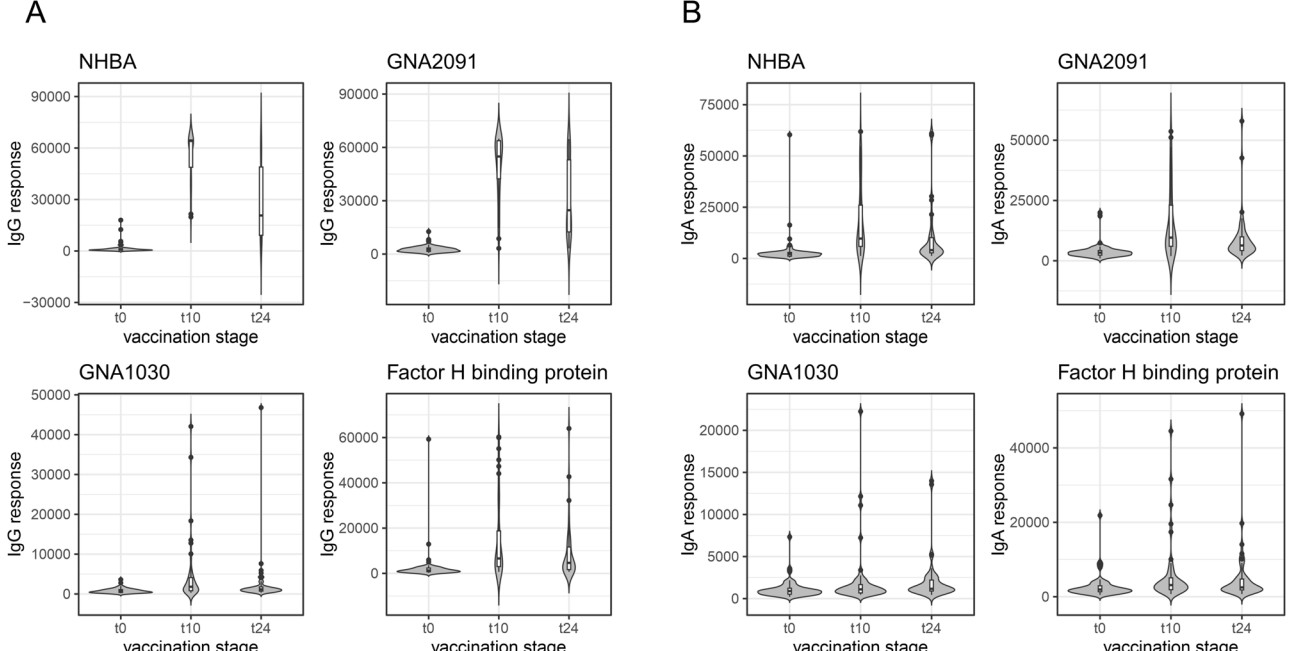

**Fig. 4 | Violin plots for IgG and IgA responses against the gonococcal variants of the recombinant protein components in 4CMenB. A** IgG responses. **B** IgA responses. Box plots are superimposed on each violin plot, providing the median, two hinges which correspond to the first and third quartiles and 'whiskers' which extend to the 'outlier' points, which are plotted explicitly. In both (**A**, **B**), n = 47, for t0, t10 and t24.

comparisons (not shown). However, the application of ASCA did separate samples from the three groups, for both IgG and IgA responses (Fig. 7). Antigens associated with these separations were compared with those identified after immunisation. Although there is some overlap, generally the antigens identified are different from those in the vaccinated cohort. Featuring strongly are several opacity protein variants (Opas), for both IgG and IgA. Some proteins

associated with iron uptake were identified, such as the TonB-dependent transporter FetA[35], the transferrin binding protein A[36] and azurin, which has been associated with resistance to oxidative stress[37]. Additionally, the IgA protease, which cleaves the host IgA and LAMP1 glycoprotein[38], was identified. Overall, these observations indicate that the profile of antigens in individuals with a Ng infection is associated with pathogenesis and survival in the host, where adhesion, iron

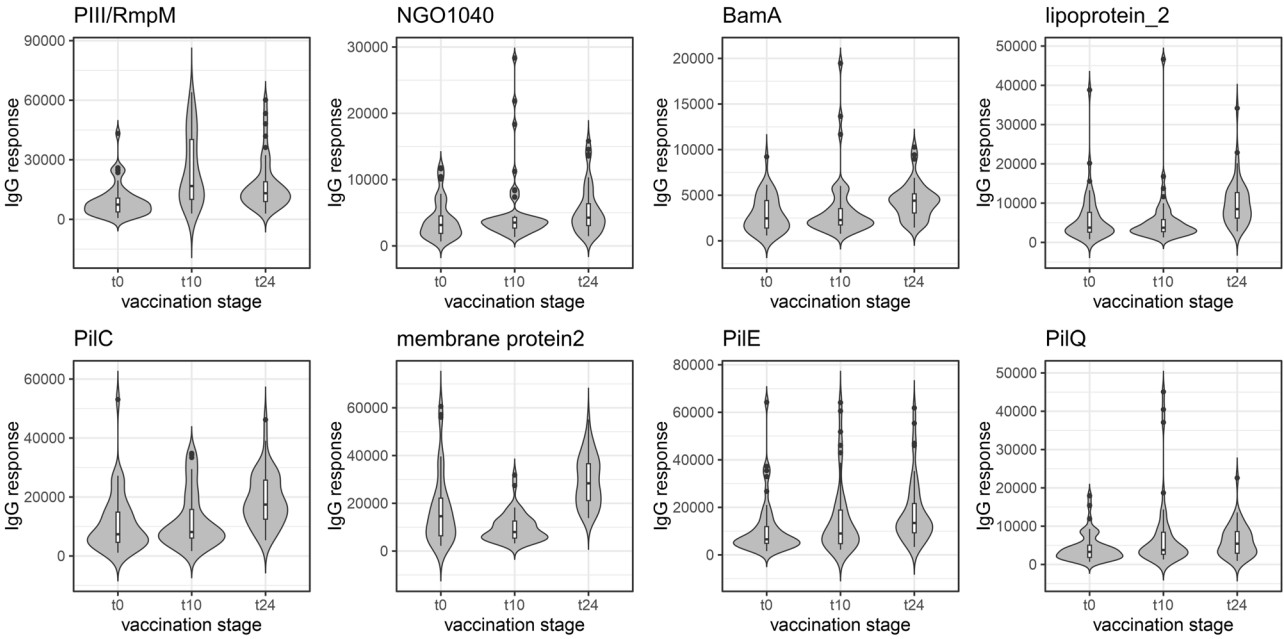

**Fig. 5 | Violin plots for IgG responses in 4CMenB-vaccinated subjects against selected gonococcal antigens.** Antigen details are given in Table 1. Box plots are superimposed on each violin plot, providing the median, two hinges which correspond to the first and third quartiles and 'whiskers' which extend to the 'outlier' points, which are plotted explicitly. $n = 47$, for t0, t10 and t24.

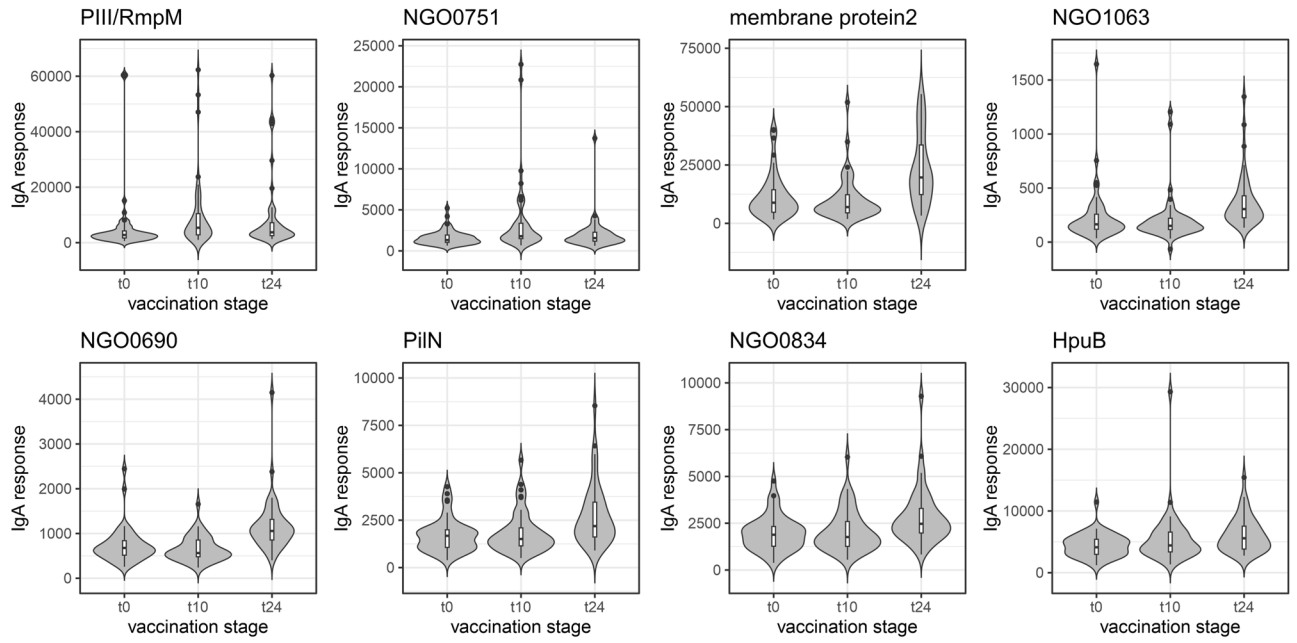

**Fig. 6 | Violin plots for IgA responses in 4CMenB-vaccinated subjects against selected gonococcal antigens.** Antigen details are given in Table 2. Box plots are superimposed on each violin plot, providing the median, two hinges which correspond to the first and third quartiles and 'whiskers' which extend to the 'outlier' points, which are plotted explicitly. $n = 47$, for t0, t10 and t24.

acquisition and resistance to host defence mechanisms play more important roles.

To provide a basis for comparison between the antigen reactivity profiles in both groups, the vaccinated and infection datasets were merged. Data from the t0 group, who were unvaccinated at the point of sample collection, were combined with the 'before' group within the infection cohort. The four recombinant antigens were removed from the overall dataset because the strong influence of NHBA and GNA2091 could obscure effects observed from OMV-derived antigens. The dataset was then subjected to analysis by ASCA (Fig. 8). Separation of the before/during/after groups was weak but, in comparison, the effect of vaccination was evident, particularly in the transition to the t10 timepoint. The t24 timepoint, however, is much closer to the after group. There are caveats to this analysis, given that the samples from the historical cohort were collected over about 10 years, prior to the vaccination trial. However, it illustrates how vaccination with an Nm vaccine induces IgG responses against a range of antigens that are different from those induced by infection.

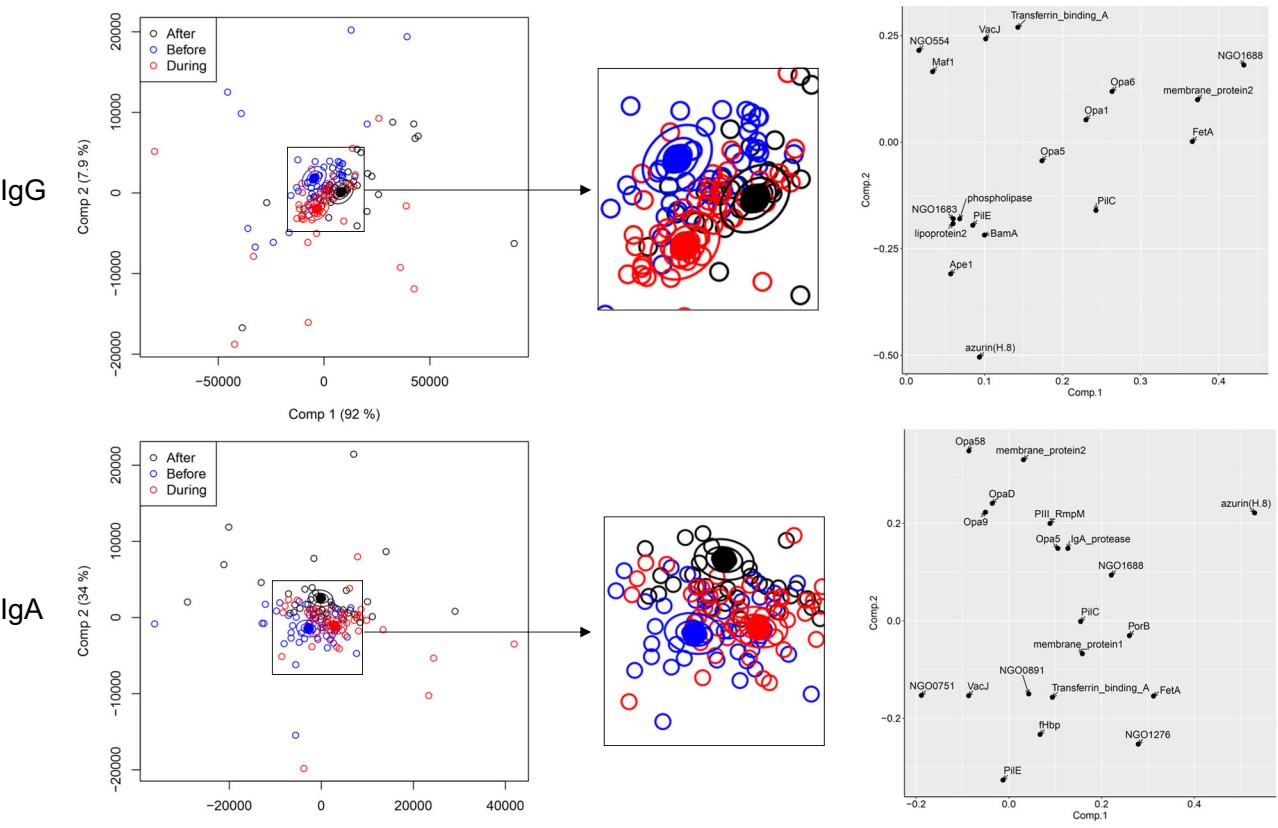

**Fig. 7 | ASCA separation of antibody profiles in non-vaccinated individuals with a Ng infection.** Layout and details are as for Figs. 1, 2, with a central magnified panel inserted. Comp 1 and Comp 2 represent the first and second principal components in the ASCA analysis. Confidence ellipsoids are plotted that reflect the estimated variation of the effect level means at 40, 68 and 95%[22]. Open symbols represent individual serum samples; ellipsoid centres are plotted as filled symbols. Righthand panels show the antigens with the highest loadings for each component.

In particular, the observation that the vaccine-induced IgG responses at the t24 timepoint have returned to a state that is closer to convalescent individuals, merits further investigation. The IgA data were less informative, although the t24 grouping was distinct from the other four groups (Fig. 8).

## Discussion

The aim of this study was to compare antibody responses in vaccinated individuals with those in individuals with Ng infection in populations at high risk of gonorrhoea. Although gonococcal infection does not usually induce protective immunity, we reasoned that analysis of serological responses in a high-risk population following immunisation with an Nm vaccine or infection would be more likely to reveal protective antigens than studies in naïve populations.

Antigen microarrays are a powerful means to dissect antibody responses to multi-component vaccines. In this investigation, we were also able to compare these responses directly with a cohort of individuals from a similar population. The datasets associated with antigen microarrays contain information about the magnitudes of responses to each individual antigen, aligned with metadata, including vaccination status, HIV infection, and gender. These are, therefore, complex, multivariate datasets which are similar in structure to those used in metabolic studies from diverse populations[21]. Treating datasets in their entirety readily distinguished the unvaccinated t0, and vaccinated t10 and t24 groups, even after removal of the recombinant antigens (Figs. 1, 2). We were, therefore, able to examine the combined effects of the OMV antigens and show that both IgG and IgA responses persist at the t24 timepoint. We found that the vaccination trial is sufficiently well controlled to support a comprehensive statistical analysis which

identified, in the case of the IgG t24 timepoint, a total of 25 immunogenic antigens. As there were fewer than 50 participants in the trial, this demonstrates the remarkable sensitivity of this approach, given that a stringent criterion ($p < 0.01$) was applied for inclusion in Tables 1, 2. Part of the reason is that the microarray scanner is capable of accurate measurement over several orders of magnitude in intensity, so even antigens that elicit low overall antibody responses are captured. This is important because it allows for sensitive detection of immunogenic antigens, even if they are present in low abundance in the vaccine or infecting bacteria. It should be noted that antigen coverage in a protein microarray is limited by antigen selection and preservation of conformational epitopes after purification and printing. The method should, therefore, be thought of as sampling a polyclonal antibody response to a range of antigens.

In this study antibody responses to 4CMenB, NHBA and GNA2091 were predominant in the vaccinated cohort, with responses to GNA1030 and fHbp also observed, but to a lesser extent. This observation agrees broadly with the findings of Semchenko et al., who reported antibodies to NHBA, GNA2091, and GNA1030 in human sera post-vaccination with 4CMenB[16]. Of these, only NHBA is thought to be exposed on the surface of Ng. For the OMV proteins, we identified a combination of integral membrane proteins, membrane-associated proteins (e.g. lipoproteins) and others as highly immunogenic antigens. The list includes proteins of a wide variety of sizes and functions (Tables 1, 2), including transport, type IV pilus assembly and adhesion. In the case of protection against meningococcal infection, the contribution of the OMV antigen components in 4CMenB is unclear. In a previous study[17,18], we reported the use of a Nm, rather than Ng, antigen microarray to measure the IgG responses in a clinical trial induced

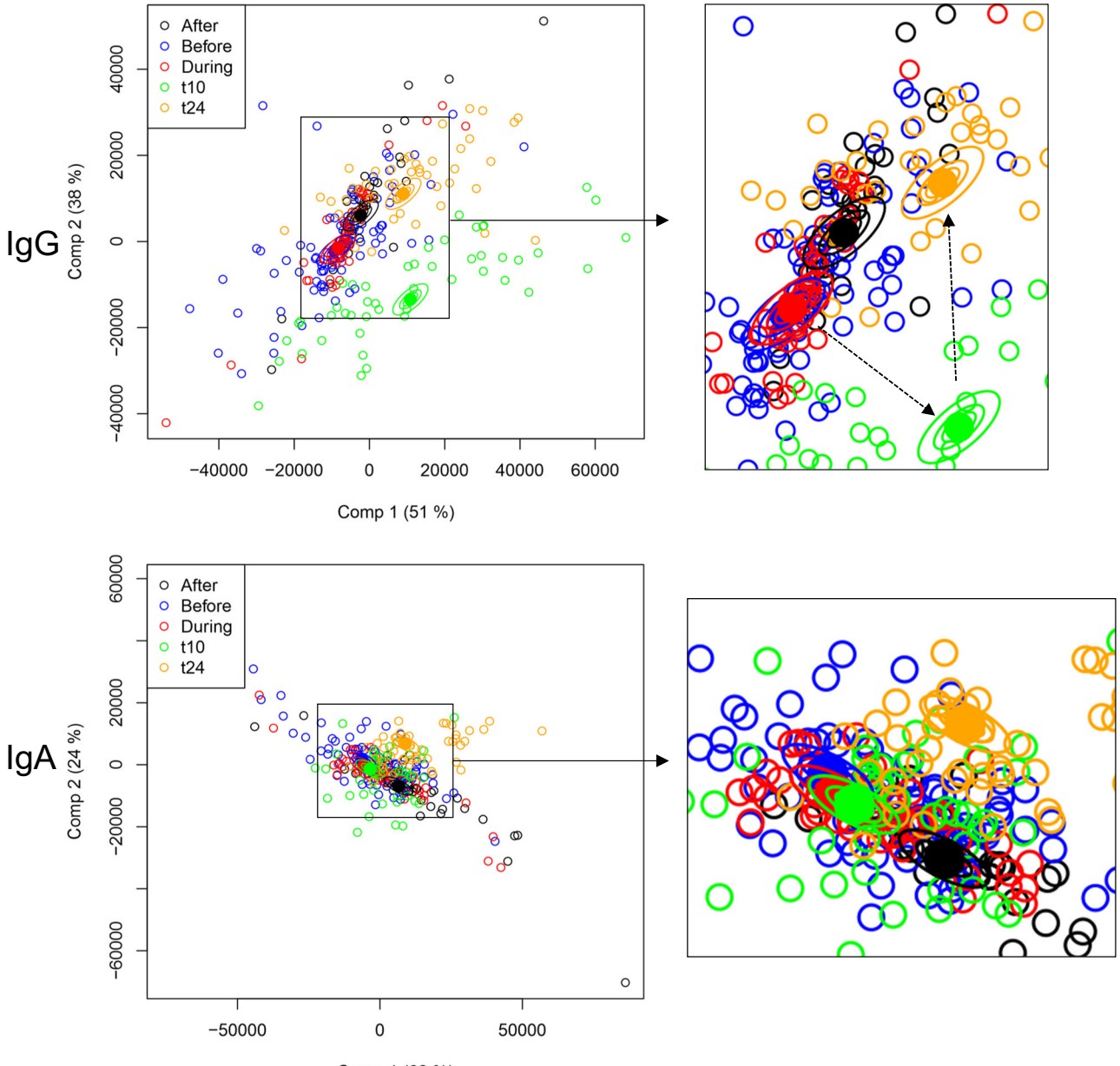

**Fig. 8 | Comparison of IgG and IgA antigen reactivity profiles in vaccinated individuals and individuals from the historical cohort.** The vaccine and infection cohort datasets were merged, antigens from the four recombinant antigens were removed and either IgG or IgA reactivities were separated by ASCA analysis, as in Figs. 1, 2, 7. Factors used were Ng infection or vaccine recipient identity and Before/During/After/t10/t24. The t0 dataset from the vaccinated cohort was merged with the 'Before' dataset in the Ng infection cohort. Layout and details are as for these previous figures, with a central magnified panel inserted. Comp 1 and Comp 2 represent the first and second principal components in the ASCA analysis. Confidence ellipsoids are plotted that reflect the estimated variation of the effect level means at 40, 68 and 95%[22]. Open symbols represent individual serum samples; ellipsoid centres are plotted as filled symbols. Arrows in the magnified IgG panel indicate transitions between before to t10 and t10 to t24.

by an OMV vaccine generated from a genetically modified Nm strain (H44/76), with constitutive expression of the iron transporter FetA[39]. We found that a regression model which included RmpM, the OpcA adhesin, the PilQ secretin, BamA and a lipoprotein (GNA1162) improved fitting to serum bactericidal data[18]. It is plausible that protection afforded by 4CMenB against Ng infection could also be attributable, at least in part, to a combination of these OMV-derived antigens. A recognised limitation of the current study is the measurement of serum, rather than mucosal IgG and IgA antibodies. Leduc et al. noted that vaginal IgG was elevated in vaginal washes from 4CMenB-immunised mice, when administered intraperitoneally, compared to alum controls[12]. Further work could explore correlations

between IgG and IgA reactivities against specific antigens and protection against infection following the challenge.

Several of the proteins listed in Tables 1, 2 have also been identified by other investigators as potential vaccine antigens, using different approaches. For example, Zhu et al. used a reverse vaccinology-type approach to identify potential Ng vaccine candidates, six of which were taken through to vaccination trials in mice[40]. Four proteins from the BAM complex- BamA, C and D- feature in our lists and a BamC homologue was also identified by Zhu et al. These authors also identified the protein referred to here as 'membrane protein 2' which is a lipoprotein of unknown function. Interestingly, this protein is recognised by both IgG and IgA antibodies, as well as

being identified by ASCA analysis of the Ng infection cohort (Fig. 7), and justifies further investigation.

It was inevitably more challenging to differentiate between the 'before', 'during' and 'after infection subgroups, given the diversity of infecting strains and differences in the timing of serum collection. Previous studies have found that humoral responses against gonococcal antigens in infected individuals are weak[41,42]; a variety of mechanisms employed by the gonococcus to suppress host immunity have been documented[43]. Our statistical analysis did not identify antibody responses to any individual antigens when judged by the same criterion ($p < 0.01$) or, indeed, with a less stringent value of $p < 0.05$. A limitation of this approach is that it considers each antigen subset individually, rather than analysing the whole dataset. We, therefore, employed a multivariate statistical package, ASCA, which combines traditional principal component analysis with ANOVA[21]. Given the extent of variability between individuals, it is encouraging that we can separate the three classes and thus extract some information about the antigens that contribute to this differentiation (Fig. 7). Although this observation will require validation with further studies, the results indicate that antibodies are raised during infection against proteins associated with pathogenesis, notably adhesion, iron uptake and resistance to host defences such as oxidative stress. The same method also enabled us to compare the specificity of IgG and IgA responses between individuals and vaccinated individuals (Fig. 8). The 'before', 'during' and 'after' categories are tightly grouped, while vaccination drives an IgG response away from the infection grouping at t10 but the t24 timepoint is closer to the infection group, although still distinct. This suggests residual IgG persists against Ng OMV antigens at least 6 months after vaccination; it would be valuable to know if this is retained over a longer timeperiod.

Studies into cellular immunity as a consequence of Ng infection have been limited to date, with some reports showing that Ng infection leads to the development of cellular immune responses[44–46] while, on the contrary, low CD8+ responses were observed in infected female sex workers[47]. These observations prompted us to investigate whether any differences could be detected in IFN-γ production from T-cells by ELISpot, using two pools of peptides identified from T-cell epitope prediction of prominent antigens that are conserved between Nm and Ng. Although we were not able to observe any statistically significant differences compared with controls, the role of cellular immunity in response to both OMV-based vaccines and Ng infection merits further study.

A retrospective case-control study by Petousis-Harris et al. provided the first evidence for a reduction in the incidence of gonorrhoea in young adults in New Zealand who had received three doses of an OMV-based Nm vaccine[8]. Although the estimated vaccine effectiveness was low (31%), this observation has been verified by several subsequent studies[48–51]. A study in South Australia reported a vaccine effectiveness of 34.7% for those who had received two doses of 4CMenB- which also contains an Nm OMV component- compared with unvaccinated individuals[49]. In this case, two doses of 4CMenB were sufficient to provide a similar level of protection to that reported by ref. 8. Abara et al. examined the incidence of gonorrhoea and chlamydia infections in young adults in New York City and Philadelphia: reduced rates of gonorrhoea were observed assuming protection for both 6 and 12 months following vaccination with 4CMenB[48]. Our observations suggest that IgG and IgA antibodies against the recombinant antigens in 4CMenB are in decline by the 24-week period and, we would presume, continue to fall after that timepoint (Fig. 4). Conversely, antibodies against many OMV antigens are increased at 24 weeks (Figs. 5, 6). It is not clear what happens over a longer period, but it is likely that several OMV antigens could be involved in mediating protection over these timescales.

There are important ramifications from our study for Ng vaccine design. The challenges inherent in Ng vaccine development are well recognised[52]: one major problem is that native immune responses to genital tract infection by Ng are not protective[53]. The limited protective effect of 4CMenB against gonorrhoea is a valuable clue, implying that cross-protection induced through meningococcal antigens, combined with the adjuvant properties of the OMV itself, could be responsible. Only NHBA, of the four recombinant antigens which we examined, is surface-exposed[16] and is not likely to be solely responsible for protection. The obvious inference is that the antigens in Tables 1, 2 are worthy of further scrutiny as potential vaccine components. In addition, the more durable responses to the OMV antigens suggest that the environment (e.g. the presence of LPS) has an adjuvant-like effect which could lead to longer-lasting responses. It is also interesting to note that antibody responses to certain antigens correlate- we have noted this phenomenon previously, even between proteins with no obvious structural or sequence homology[18]. These observations suggest that devising a vaccine against gonorrhoea does not merely require the selection of component antigens and adjuvants, but a consideration of how the whole formulation is able to induce long-lasting, protective immunity.

## Methods
### 4CMenB trial
The clinical trial complies with all relevant ethical regulations and was approved by the KEMRI Scientific and Ethical Review Unit (approval number: CSC 182) and the University of Oxford (approval number: 16-20). Written informed consent was obtained from all study participants. The trial was conducted between June 2021 and February 2022 (clinicaltrials.gov identification: NCT04297436) using Bexsero, a licensed Nm vaccine, in 50 participants aged 18 to 25 years who attended the Kenya Medical Research Institute (KEMRI) clinics in Mtwapa and Malindi, Kenya. The KEMRI cohort studies in Mtwapa and Malindi are supported by a 'key populations study' community advisory board (CAB). The CAB consists of stakeholders from LGBTQ organisations, business leaders, bar owners, religious leaders, community security, chiefs, police, village elders, as well as lawyers, human -rights activists, and leaders from several LGBTQ CBOs along the Kenyan coast. A copy of the Study Protocol is included in the Supplementary Information file. Most participants were sex workers and/or men who have sex with men (MSM). Participants living with HIV were clinically stable, with no new WHO stage 3 or 4 opportunistic infections, and virologically suppressed. Cohort participants aged 18–25 years old were screened for eligibility and excluded from the study if they were pregnant, known to have severe allergic reactions or bleeding disorders, or living with HIV with a viral load ≥200 copies per mL. Following enrolment, participants made four visits over a 24-week study period. Bexsero was given at baseline and week 8. Serum was collected for immunology studies at three timepoints: at baseline (prior to Bexsero vaccination, t0), 2 weeks after the second Bexsero vaccination (t10), and at study completion at week 24 (t24). White blood cells were collected for ELISpot analysis at baseline and at t10. Three participants (two men and one woman) had a prior Ng isolate that has been fully characterised. These three participants were fully vaccinated.

To diagnose Ng infection, swabs of the oropharynx, urine/cervix, or anus were collected at t0, t10 and t24 for point-of-care GeneXpert® CT/NG assay (Cepheid AB, Sweden). Participants with an Xpert-diagnosed CT or Ng infection received appropriate treatment. Swabs were taken from Xpert-diagnosed Ng participants and inoculated on Thayer Martin Modified agar; plates were transported on the same day to a laboratory for culture and antimicrobial susceptibility testing.

### Historical cohort of individuals with laboratory-confirmed gonococcal infection
Between June 2010 and July 2019, 85 cohort participants (18–49 years) were identified who had a laboratory-confirmed gonococcal infection. Participants included sex workers and MSM attending the KEMRI

clinics in Mtwapa and Malindi[54]. Ethical approval for the analysis of gonococcal isolates and plasma was granted by the KEMRI Scientific and Ethical Review Unit (approval number: 2842). Samples were obtained by swabbing and screening for Ng by Gram stain, oxidase test, and API-NH (bioMerieux, France)[54]. Routinely, volunteers who reported rectal anal intercourse (RAI) were offered proctoscopy[55]. Since 2016, participants who reported receptive anal intercourse had a swab collected for GeneXpert® CT/NG assay (Cepheid AB, Sweden)[56].

Ng infection was diagnosed in men with urethral or rectal discharge, in men who reported RAI, and in women irrespective of symptoms. Of 85 individuals with a laboratory-confirmed Ng infection, 42 were microbiologically confirmed, three by Xpert confirmation only, 34 by Gram stain confirmation only, and three by GeneXpert and Gram staining. Plasma samples were collected at the following time-points: (i) before 1–6 months before Ng infection, (ii) during at the date of the Ng infection visit to up to 4 weeks after diagnosis was confirmed and (iii) after 4 weeks–3 months after Ng diagnosis for participants who were in the HIV negative or acute HIV infection cohort study[57]. HIV-positive participants had an enrolment plasma sample collected only. Paired plasma samples from 17 individuals were obtained before, during and after Ng infection. In addition, 22 participants had plasma samples collected before and during Ng infection, 14 during and after, 8 before and after, 7 before infection only, 11 during infection only and 6 after infection only. 'Before' samples were obtained for a median period of 31 days (range 27–48) pre-infection. 'After' samples were obtained from individuals positive for Ng infection for a median period of 31 days (range 29–60) post-infection.

### Antigen panel
The proteome of the Ng strain FA1090 was accessed from the NCBI database and candidate antigens were screened systematically using several bioinformatic tools. First, transmembrane beta-barrel outer membrane proteins were predicted using PRED-TMBB2[58] and BOMP[59]. Proteins were included in the final selection if the results of at least one beta-barrel prediction tool were positive. Secondly, the proteome was uploaded to the BUSCA[60], PSORT and CELLO2GO bioinformatic tools to identify the predicted subcellular location of the gonococcal proteins. Proteins with predicted cytoplasmic location were excluded. SignalP 5.0 tools[61] was then used to predict gonococcal proteins containing N-terminal signal peptides. Proteins identified by all three tools (BUSCA, PSORT and CELLO2GO) to be of non-cytoplasmic origin and containing a signal peptide at their N-termini were included in the final selection, alongside predicted beta-barrel proteins. The FA1090 gonococcal strain lacks the expression of outer membrane lactoferrin-binding proteins A and B, so these antigens were included in the final selection based on the amino acid sequence of gonococcal strains NG104 and NG102, respectively. Additionally, major outer membrane porin proteins (PorB) from eight different alleles were also included in the final antigen selection. A summary of the selected proteins is provided in Supplementary Data 1.

### Cloning, expression and purification of gonococcal antigens
The open reading frames (ORFs) of selected antigens were cloned into pET30a(+) (Novagen) using either the high-throughput In-Fusion™ cloning method or restriction cloning. All recombinant plasmids generated were sequenced (Eurofins or Genewiz) and confirmed with reference to the FA1090 gonococcal genome from the nucleotide database at NCBI. Recombinant plasmids were transformed into XL-10 Gold ultra-competent Escherichia coli (Stratagene). Plasmids encoding insoluble proteins were subsequently transformed into Lemo21 (DE3) competent E. coli (New England Biolabs) for protein expression. Cultured cells were grown in MagicMedia™ (ThermoFisher) for 24 h at 37 °C with shaking at 200 rpm. To ensure correct formation of disulphide bonds, soluble proteins were transformed into SHuffleT7 K12 E. coli cells (New England Biolabs) and grown in MagicMedia

(ThermoFisher Scientific) for 24 h at 30 °C with shaking at 200 rpm. Antibiotic selection was maintained using 50 μg/mL kanamycin. The recombinant proteins were purified following the protocol from ref. 17. Insoluble proteins were extracted and refolded from inclusion bodies whilst soluble proteins were purified directly with no further processing. All solubilised target proteins were subject to sequential purification steps by immobilised metal affinity chromatography (HisTrap HP 5 mL column; Cytiva) and size exclusion chromatography (Superdex S-75 or S-200 Hi-Load 16/600 column; GE Healthcare) on an AKTA-FPLC system (GE Healthcare). The oligomeric state of the proteins was determined during this chromatographic step. Eluted protein samples were analysed for purity by SDS-PAGE and concentrated with Vivaspin® centrifugal concentrators (Sartorius). All purified proteins were concentrated to 1 mg/mL; protein concentration was estimated from the absorbance at 280 nm and stored at −80 °C prior to microarray printing.

### Antigen microarray fabrication
Gonococcal protein microarray slides were custom-made by Arrayjet Limited, UK. In total, 91 recombinant gonococcal protein samples were printed onto microarrays after dilution to 0.5 mg/mL (1:1 ratio) with JetStar printing buffer® (Arrayjet). Three sets of control samples were used for the slides, consisting of monoclonal mouse and human IgG. Mouse and human IgGs were prepared at eight concentrations, with the highest concentration of 0.5 mg/mL and serially diluted with JetStar printing buffer® (Arrayjet) down to 0.0035 mg/mL. Each microarray slide consisted of 16 identical blocks, with each mini-array containing 91 gonococcal protein preparations and 17 control samples printed in five repeats. The slides were stored at 4 °C until use.

### Validation of protein spots on microarray slide
Protein spots were assessed for quality, circularity, and consistency using mouse monoclonal anti-tetra-histidine IgG antibody (Qiagen) diluted 1:2000 in TBS (20 mM TrisHCl, 150 mM NaCl, pH 7.6) to detect hexa-histidine-tagged antigens. An anti-mouse IgG antibody (Sigma) diluted 1:5000 in TBS was used to detect bound antibodies. Signal intensities were quantified and normalised against the background using the Mapix v9.1.0 Microarray Image Analysis software (Innopsys).

### Microarray immunogenicity probing
Slides were blocked with 200 μL of SuperG™ Blocking Buffer (Grace Biolabs, US) per well and incubated for one hour at room temperature (RT). From previous experience in the laboratory, optimal dilutions of sera could range from 1:50 up to 1:1000; for these samples, a 1:300 dilution was selected as showing the best balance between foreground and background signals. Sera were diluted 1:300 with TBS, 100 μL were added to each mini-array and incubated at 20 °C for 1 h. After washing three times with 300 μL TBS-T (0.05% Tween-20) and once in TBS for 10 min, slides were incubated in the dark for 1 h at 20×°C with 100 μL of Goat Anti-Human IgG Fc (DyLight® 650) pre-adsorbed secondary antibody (ab98622, Abcam, UK), using a working dilution of 1:5000 in blocking agent. The arrays were incubated for IgA detection with 100 μL of Goat Anti-Human IgA alpha chain (DyLight® 650) pre-adsorbed secondary antibody (ab96998, Abcam, UK), using a working dilution of 1:5000 in blocking agent. Slides were then washed three times with 300 μL TBS-T (0.05% Tween-20) and once in TBS for 10 min. Slides were rinsed in de-ionised water and dried by centrifugation at 200×g for 2 min. They were scanned in an InnoScan 710 (Innopsys, France) with the photomultiplier tube set to 40% for 635 nm. Image analysis and data quantification was carried out using Mapix v9.1.0− Microarray image acquisition and analysis software (Innopsys, France).

### Microarray data acquisition
Microarray spot intensities were quantified using Mapix v9.1.0 Microarray Image Analysis software (Innopsys, France), which utilises

automatic background subtraction for each spot. The spot intensities for each protein were recorded in quintuplicate; arithmetic means were determined, and spot intensities for buffer-only controls were subtracted.

## Bactericidal methods

Serum bactericidal activity against Ng strain F62 was determined for each serum sample (t0, t10, t24) from each subject from the Bexsero trial as described[12], with the exception that IgG- and IgM-depleted pooled normal human serum (NHS) (PelFreeze) was used as the complement source. All three serum samples from each subject were tested in the same microtiter plate to minimise technical differences in the assay that might influence the comparison of the titres. The antiserum dilution that gave 50% recovery compared to wells without antiserum was defined as the bactericidal50 titre. Wells containing heat-inactivated NHS and a 1:960 dilution of each serum sample were tested in parallel to measure the complement-independent loss of bacterial viability during the assay; no appreciable loss was detected in any experiment.

## T-cell epitope prediction and ELISpot

An informatics pipeline was initiated to identify the most likely high-affinity T-cell epitopes within the major antigens present in 4CMenB. A total of 22 antigens, identified by ref. 62, were screened using the T-cell epitope prediction tool for binding to MHC-II at the IEDB Analysis Resource (www.iedb.org)[63,64]. The recombinant antigen NHBA was also included in the analysis. The class II alleles used for the search were matched to those found to be most common in a similar Kenyan population[65]: DQA1*01:02/DQB1*03:01 and DPA1*01:03/DPB1*01:01. The top 20% in the hits ranking were sorted, overlapping peptides eliminated and sequences extracted with a high level of conservation between the meningococcal sequence and a selection of gonococcal sequences from diverse strains. The highest-ranking eight peptides were divided into two pools, to reduce the number of ELISpot assays required; details are given in Table S1. A separate pool of peptides was prepared containing the same composition, but with the sequences scrambled (Table S1). These were used as a comparison group for statistical analysis, which was conducted using the Wilcoxon paired rank test.

## ELISPOT assay

An IFNγ ELISpot assay was used to quantify the frequency of antigen-specific effector T-cells producing IFNγ after overnight culture. In brief, ELISpot 96-well plates (Millipore, UK) were coated at 10 µg/ml with a monoclonal antibody against IFNγ (Mabtech, AB, Sweden) overnight. To each well, 200,000 cryopreserved PBMCs, isolated as described by ref. 66, were incubated for 16–18 h of stimulation in the presence of peptide pools (10 µg/ml) in triplicate wells. PBMCs had a mean viability of 97%. After incubation, PBMCs were discarded, the plate was washed and secreted IFNγ was detected by adding 1 µg/ml anti-IFNγ biotinylated monoclonal antibody (Mabtech) for 4 h at room temperature (RT). The plate was then washed, 1 µg/ml streptavidin alkaline phosphatase was added for 2 h at RT, and the plate was washed again before adding an alkaline phosphatase substrate. The substrate was left to develop until spots were clearly visible. After that, the plate was washed. Following overnight drying, the spots in each well were enumerated by an automated plate counter.

The two pools of peptides generated from the eight highest-ranking peptides described above were included as test conditions. Two positive controls were employed a) phytohemagglutinin-L (PHA-L, Sigma), used to induce a mitogenic response and demonstrate viability of lymphocytes b) human anti-CD3 antibody as a polyclonal T-cell activator. A negative (no peptide) control was included to detect non-specific IFNγ release, and any response in the negative control were subtracted from the antigen-specific responses.

Counts were presented as IFNγ-Spot-Forming Units per $10^6$ PBMC (SFU), which is determined by taking an average of the triplicate wells and then subtracting any background response (defined as the response to media alone without antigen stimulation). Statistical analyses were performed in GraphPad prism version 8.4.1. The Wilcoxon matched-pairs test was used to compare pre- and post-vaccine responses, with $p < 0.05$ considered significant.

## Statistics and reproducibility

The sample size for the study was guided by our previous experience with a similar study which we conducted using a meningococcal microarray[17]. The sample size in this study is about twice the number of participants: this was motivated by the different nature of the study, which was looking for cross-reactive antigens and may, therefore, have been less sensitive. Technical improvements optimised the sensitivity of this study: over 20 unique antigens were identified in the vaccine study using a stringent $p$ value cutoff of 0.01 (Tables 1, 2). Numbers for the historical cohort study were roughly equivalent, allowing a direct comparison between the two groups. Despite the variation in the latter group, some separation by disease status was possible (Fig. 7). No data were excluded from the analyses. Laboratory workers engaged in microarray data collection were blinded to sample identities. Each array contained a series of antibody concentrations, which were used as a standard to ensure reproducibility between slides and verify linearity. In addition, the reproducibility of the antigen profiling data was directly tested on a randomly chosen subset of serum samples for all the proteins contained in the microarray. Correlation R coefficients were obtained with values between 0.7 and 0.8. All computation on the microarray data, including the Wilcoxon paired rank test version 3.6.2, was carried out using dedicated scripts in R version 4.3.2. An approximation is used to calculate the $p$ value in the Wilcoxon paired rank test analysis; $p$ values were adjusted to correct for multiple comparisons[67]. ANOVA simultaneous component analysis (ASCA) was used to conduct regression analysis for dependent variables (antibody reactivities) with multiple factor variables. The latter consisted of two categorical variables: vaccine recipient identity, which was arbitrarily assigned, and the vaccination stage (t0, t10, t24). ASCA analysis was carried out using asca[21] as part of the multi-block suite version 0.8.7. tSNE[23] was implemented using the Rtsne package version 0.17 which uses the Barnes-Hut approximation[68]. PCA analysis used the prcomp and factoextra packages version 1.0.7, implemented in R version 4.3.3.

## Reporting summary

Further information on research design is available in the Nature Portfolio Reporting Summary linked to this article.

## Data availability

The microarray data generated in this study have been deposited in the GEO (Gene Expression Omnibus) database under accession code GSE269648 [https://www.ncbi.nlm.nih.gov/geo/].

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

## Acknowledgements

We thank the staff at the KEMRI clinic in Malindi for their support to the study, and study participants for their participation in the trial. This manuscript was submitted for publication with permission from the Director of KEMRI. This work was funded by a Wellcome Trust Collaborative Award (214374/A/18/Z). The funder played no role in the conceptualisation, design, data collection, analysis, decision to publish, or preparation of the manuscript.

## Author contributions

C.T., E.J.S., J.P.D., A.J., M.C.J.M. and I.M.F. devised and supervised the project. E.J.S. designed and conducted the vaccine clinical trial. L.S., A.T., F.R.B. and S.R. expressed and purified proteins for inclusion in the microarray. Data were collected by A.T. and F.R.B. and analysed by F.R.B. and J.P.D.. O.C., J.C., E.N. and G.B. conducted and analysed the ELISPOT assays. A.J. conducted serum bactericidal activity measurements. O.H. carried out the analysis of sequence conservation. I.M.F., G.B. and A.C. provided advice concerning the design of the study and interpretation of the results. The manuscript was written by J.P.D., F.R.B., C.T. and E.J.S., with comments and modifications from all authors. C.T. was the lead applicant on the grant that funded the work; A.J., J.P.D., M.C.J.M., I.M.F. and E.J.S. were co-applicants.

## Competing interests

JPD has been in receipt of funding from GSK to support PhD students in his laboratory. The remaining authors declare no competing interests.
