## [Peer Review File · Nature Communications]

REVIEWER COMMENTS

Reviewer #1 (Remarks to the Author):

This manuscript describes the IgG and IgA immune responses to *Neisseria gonorrhoeae* (Ng) antigens following Ng infection or vaccination with the 4CMenB (Bexsero®, GSK) vaccine that is licensed vaccine against the closely related bacterium *Neisseria meningitidis* (Nm). Overall, this research is of interest to the Ng vaccine field as there is growing interest in using 4CMenB to protect against Ng. The key results of the study are that there is increased IgG and IgA targeting Ng antigens post 4CMenB vaccination and that vaccine-induced antibody responses to Ng differ to antibody responses following infection. Key Ng antigens recognised by vaccine-induced antibodies are also identified. This study also presents a powerful method for analysis of antigen-specific immune responses.

A key question is whether the history of Ng infection in vaccinated participants known? This could significantly impact vaccine-induced immune responses. In the absence of this information, the comparison with unvaccinated but infected participants is difficult to interpret.

Specific comments/questions are as follows.

Abstract

Line 42 – “A cohort of similar patients with laboratory-confirmed gonococcal infection before, during, and after infection showed weaker and different reactivities.” This sentence needs to be rewritten for clarity.

Possibly - A cohort of similar patients with laboratory-confirmed gonococcal infection showed weaker and different reactivities before, during, and after infection.

Introduction

Line 50 – the genitourinary tract is not the only site of infection, and you should mention this, especially since you do test for anal and pharyngeal infection.

Line 65 – consider adding the vaccine name, and at line 72 adding that the OMVs are the same as in MeNZB.

Line 73- there are now several studies that report potential cross protection of 4CMenB against Ng that should be referenced.

Some details of the recombinant antigens and their surface localisation in Ng would be useful in the introduction as antibody responses to fhbp, GNA1030 and GNA2091 are all described in detail in the results. The absence of NadA in Ng could also be mentioned.

Materials and Methods

The methodology used is sound, meet the expected standards in the field and there is enough detail provided for the work to be reproduced. The methods are possibly too detailed for the main paper, and it might be good to consider reducing the level of detail if this is available in published papers (e.g., are the details of the cloning, expression, and array printing outlined in Ref 16?) or to have less details in the main paper if a more detailed methods section can be included in the supplementary data.

Line 113 – Is data available for past Ng infections in the vaccine cohort?

Line 145 – Consider changing “gonorrhoea was diagnosed” to “Ng infection was diagnosed”.

Line 148 – Was plasma or serum collected? Typically, serum is analysed in SBA assays, and you later refer to serum.

Line 188 – check typo for rpm.

Line 301 – How similar are the main antigens in strain F62 compared to FA1090 used in the arrays?

Line 330 – Can a reference be added for the collection and processing of PBMCs?

Please update ug to μg throughout.

Results

Line 377-382 and beyond – Can some of these details could be moved to the methods section. It is hard to appreciate the key results in this section due to the complexity of the methods used. Perhaps a clear result and what that means in simple terms for the immune response measure can be stated followed by how it was reached.

Line 385 – What are the points with the ellipsoids? Are these for the average of all participant responses to all antigens? Can this be clearly explained?

Line 421 – The meaning of the results regarding Opa are unclear. What is the sequence identity of Opa in the vaccine compared to the Ng sequences?

The value and importance of using Opa and PorB variants on the array is not clear. Vaccine-induced immune responses would presumably focus on the variant in the vaccine, while cross-reactivity to different Ng variants may imply ability to recognise/cross protect against different Ng strains. Are specific variants more common in Ng strains?

Line 431- Could the reactivity to different Opa variants be due to the conserved regions? If not, does this imply immune reactivity due to Ng infection rather than 4CMenB vaccination?

Table 2 – Stage – “paired t0 with t10t10 or t0 with t24t24” – what does the second t10 refer to in t10t10?

Figure 4 – it would be useful to add IgG and IgA above the graphs on the left and right, or A and B to the panels.

Line 596 – I suggest changing “patients” to “individuals with a history of Ng infection”, “infected individuals” or similar. Consistency of the terms used should be checked throughout the manuscript as this differs in Figures and text.

Line 619 – The phrase “disease-associated individuals” should be changed to something to refer to Ng infection as this is what was measured/included, regardless of disease presentation. Also, for Figs 7 and 8.

Line 621 – “are more important” – it is not clear what this refers to.

Figures 1, 2, 7 – It is hard to appreciate / compare the spread of immune responses as the y-axes are different.

Discussion

Line 674 – In your previous study, can you please outline what OMV vaccine was used, if this was in humans or animals, and how the findings of that study differed to this one.

Line 729 – This sentence is unclear. Did you investigate protection against Ng infection in your study? If so, what was the control group and the level of protection seen?

Reviewer #2 (Remarks to the Author):

In this manuscript, Stejskal et al. characterized the IgG and IgA antibody responses against *Neisseria gonorrhoeae* protein antigens in a high-risk population of people vaccinated with the 4CMenB anti-meningococcal vaccine. The authors furthermore evaluated the humoral immune responses in a patient population that had been infected with *N. gonorrhoeae*. The authors conclude that the outer membrane vesicle component of the 4CMenB vaccine elicited more durable responses than the recombinant proteins in the vaccine preparation. The authors also speculated that some of the same antigens are also immunogenic during natural infections.

1. In terms of presentation, some of the figures and table would be easier to read/interpret with more information. In Tables 1 and 2, it was not immediately clear what the horizontal line in the table represented. This could be clarified in the footnotes. Also, why is t10 or t24 duplicated in the last column? The axes in the figures 1-3 are either not labeled or not clear

what they represent. It would help to clarify what the violin plots represent, specifically what is the horizontal spread meant to convey?

2. Line 49-50: should be “sexually transmitted infection”

3. Line 50: Not clear why the authors are explicit about “infection of epithelial cells of the genitourinary tract” specifically when other sites of infection are clearly relevant, even to these studies.

4. Line 51: These statistics are pretty old now, coming from 2016. Much newer data are available.

5. Lines 54-55: Not clear what is meant by “the high incidence of multidrug antibiotic resistance has increased concern about the prevalence of Ng infection”.

6. Line 56: Why are gonorrhoea impacts on the reproductive health of women particularly a concern in Africa and not everywhere?

7. Lines 86-97: the word protective/protection is duplicative.

8. Line 124 and elsewhere: is “enrolment” spelled correctly?

9. Lines 263-264: It is not clear why EBNA-1 viral protein is a positive control.

10. Did the investigators only use a single dilution (1:300) of human serum? This could limit the analysis severely as some antigens may not be detected at this dilution.

11. A significant concern for the entire analysis is the level of sequence variation between the *N. meningitidis* vaccine antigens and the FA1090 or F62 gonococcal strains used for testing. It is not clear to what degree sequence diversity between antigens can be distinguished from degree of “antigenicity”. Throughout the presentation, the authors refer to “the magnitudes of responses to each individual antigen” (line 649) but the amount of signal is related to both the magnitude of the response and the degree of sequence conservation. Little to no attention was given to the latter.

12. In several instances (also highlighted as – in % identity tables 1 and 2) the authors indicate that antibody responses were detected against a gonococcal antigen even though no meningococcal sequence homolog was detected. It is not clear how these responses then would be generated. These observations raise some doubt about the analysis unless the authors have a reasonable explanation, which was not provided.

13. Line 469: This observation must correspond to Figure 3 (not Figure 2) as Figure 2 has no panel C.

14. Line 513: What is meant by “intrinsic and extrinsic membrane proteins” that are transmembrane beta-barrel structures?

15. Lines 514-516: What is meant by “the inclusion Indicates the importance of outer membrane protein folding and insertion”?

16. Speaking of membrane folding, how can it be ensured that the purified proteins in the microarray are correctly folded? If this cannot be assured, then it is likely that many antibodies against conformational epitopes have been missed in the analysis.

17. Line 520: Because pilE is subject to high-frequency antigenic variation, the sequence identity (listed as 66%) is not fixed when comparing the 4CMenB strain and FA1090.

18. Given the technical issues with the ELISpot analysis, the data seem premature for this presentation.

19. Line 616: HpuB does not seem to be represented in Figure 7, unlike what is indicated in the text.

20. With respect to the natural infection study, how can the impact of multiple previous infections be taken into account in terms of antibody “levels” detected? Also, this analysis is even more problematic with respect to the divergence of infecting strains, their sequence conservation with each other and with the type strain used here (FA1090).

Reviewer #3 (Remarks to the Author):

The study Rationale and objectives are to use the meningococcal B vaccine (4CMen B/Bexsero) as an immunological probe to identify *N. gonorrhoeae* (Ng) antigens by cross-reactive responses in individuals with high Ng exposure as recurrent exposure to Ng may elicit some immunologic memory to support Ng vaccine development. The primary objective is to assess if immunisation of individuals at risk for gonococcal infection with 4CMenB (Bexsero) elicits humoral and T cell cross-reactive responses against Ng.

The design is an open label, single arm study of 2 doses of Bexsero, which contains the MeNZB OMV component plus three recombinant antigens (NadA, fHBPGNA2091, and NHBA-GNA1030); there is a high level of sequence identity exists between MeNZB OMV (the vaccine and Bexsero OMV antigens, and gonococcal proteins. NHBA is the only 4CMenB vaccine (Bexsero®) recombinant antigen that is conserved and surfaced exposed in Ng.

The study population is a cohort of MSM and FSWs in coastal Kenya which has a high incidence of GC infection: incidence estimates for Ng (based on Xpert testing) in HIV-negative 18-25 year-old MSM participants is 21.4 per 100 person-years (95% CI: 13.1-35.0), and 43.5 per 100 person-years (95% CI: 23.4-80.8) in HIV-infected individuals

Immune response methods: Comparison of the pre/post immunization sera will provide a comprehensive overview of Bexsero antigens which induce Ng-cross reactive IgG responses and could therefore be responsible for Nm OMV-mediated protection against Ng. Samples with cross-reactive antibodies against the OMV antigens, which may offer protection against Ng will be selected to inform antigen choice and vaccine design.

Humoral responses: serum antigen- and OMV-specific IgG1, IgG2a, Ig3A and IgA titres will be determined by ELISA approximately two weeks after final immunizations. ELISA assay using peptides will covering recombinant protein antigens, purified antigens, and defined OMVs Responses will be compared among pre-post test groups by repeated measures.

Th1 responses: all candidate vaccines that show efficacy in the mouse models against Ng (2C7 peptide

mimetic, Gc OMVs administered with IL-12) induce Th1 responses.

Comments for the authors:

Abstract: 1) “add to protect against infection and reproductive health consequences” to “driven by antimicrobial resistance, 2) add ‘single arm open label study of 2 doses of Bexsero

and humoral and cellular immune response was measured at 4 time points over six months in which pre-post immunization time points were compared by a T test". 3) Clarify "serum IgG and IgA reactivities against the gonococcal homologs of the recombinant antigens in the vaccine peaked at 10 weeks but declined by 24 weeks. The reverse was the case for most antigens originating from the OMV component" and why that leads to their conclusion that cross-protection of the 4CMenB vaccine against gonorrhoea could be explained by cross-reaction against a diverse selection of antigens derived from the OMV component."

Introduction:

1) Lines 64-66: would clarify that this was not a randomized evaluation of Bexsero and rather a retrospective observational cohort finding.

Methods:

1) Lines 146-148: In the historical cohort, there were 85 NG infections. Please indicate how many of the 50 participants had a prevalent Ng infection at baseline and how many were negative at baseline and had 1 or more incident Ng infections?

2) Lines 161: How closely related is the gonococcal strain FA1090 to circulating Ng strains in Kenya?

3) Lines 277-292: Were paired plasma samples analyzed on the same microarray slides to minimize variability in immune response due the amount of protein on different slides?

4) Line 302: Why was Ng strain F62 selected for the SBA? Could a panel of Ng strains been evaluated for SBA?

5) Justify the 6 months follow-up and whether that is sufficient to assess chronology and durability of different immune responses.

6) Given the potential that HIV infection could confound the immune response, justify including both persons living with HIV and HIV uninfected: Should the results be analyzed separately?-

7) Given that this is a highly exposed cohort, did they stratify by whether they had NG immune response at baseline?

8) Did they do a sensitivity analysis in which they excluded those who acquired CT during follow-up as that could have modified their immune response to Ng (based on observational data from New Zealand)?

9) Did the investigators adjust for multiple comparisons given the large number of antigens assessed?

Results:

10) Lines 367-8: Clarify whether there was only a single Ng infection during follow-up (e.g., any infections earlier than the one described at month 6)?

11) Lines 546-557: Did you assess the viability of the PBMCs, given that collection, storage and shipping could have reduced their viability?

Discussion:

13) Lines 406-418 and Fig 3 ABC: Discuss whether the different IgG and IgA reactivities, the higher IgG than IgA reactivity at t24, and the highest reactivity to NHBA and GNA2091 was also seen in your study of Bexsero in the lower risk cohort in the UK. This would be of interest in

terms of whether this is a consistent finding in a cohort with lower risk of previous exposure to Ng.

14) Lines 541-2: What are the implications of lack of evidence of IgA or IgG responses to individual antigens and SBA?

15) Line 648: Clarify that you are comparing the Bexsero responses to a similar Kenyan population who had Ng infection and samples obtained during and after acquired infection.

16) Lines 743-745: Can you propose criteria for prioritizing the different antigens in Tables 1 and 2: strength of response, durability at T10 and T24, and/or correlation between IgG responses to different antigens which have been observed in this and prior studies of humans immunized with Bexsero? What are the next steps?

We thank the reviewers for their valuable comments on the manuscript.

Reviewer #1 (Remarks to the Author):

This manuscript describes the IgG and IgA immune responses to *Neisseria gonorrhoeae* (Ng) antigens following Ng infection or vaccination with the 4CMenB (Bexsero®, GSK) vaccine that is licensed vaccine against the closely related bacterium *Neisseria meningitidis* (Nm). Overall, this research is of interest to the Ng vaccine field as there is growing interest in using 4CMenB to protect against Ng. The key results of the study are that there is increased IgG and IgA targeting Ng antigens post 4CMenB vaccination and that vaccine-induced antibody responses to Ng differ to antibody responses following infection. Key Ng antigens recognised by vaccine-induced antibodies are also identified. This study also presents a powerful method for analysis of antigen-specific immune responses.

A key question is whether the history of Ng infection in vaccinated participants known? This could significantly impact vaccine-induced immune responses. In the absence of this information, the comparison with unvaccinated but infected participants is difficult to interpret.

Only 3 of the 50 participants in the BX trial had a laboratory-confirmed Ng infection. Our experience in this and previous studies (Awanye et al. reference 17) is that background IgG immunoprofiles in any particular individual are remarkably stable- an individual will have a pattern of responses to particular antigens which persist during the vaccination trial (this observation is readily apparent from a visual inspection of the heatmaps in Fig S1). The comparison to which the reviewer refers (Figure 8) merges the t0 group, from the vaccine study, and the 'Before' group, from the disease cohort, to compare the relative stimulation in antigen responses during vaccination or disease. This is a valuable insight, because it compares antibody responses between vaccine recipients and individuals during infection. We have revised the text in the Results section to make this point more clearly, as well as acknowledge the limitations of the approach (lines 524 to 538).

Specific comments/questions are as follows.

Abstract

Line 42 – “A cohort of similar patients with laboratory-confirmed gonococcal infection before, during, and after infection showed weaker and different reactivities.” This sentence needs to be rewritten for clarity.

Possibly - A cohort of similar patients with laboratory-confirmed gonococcal infection showed weaker and different reactivities before, during, and after infection.

Now changed to 'A cohort of similar individuals with laboratory-confirmed gonococcal infection were compared before, during, and after infection: their reactivities were weaker and differed from the vaccinated cohort.' (lines 15 to 16)

Introduction

Line 50 – the genitourinary tract is not the only site of infection, and you should mention this, especially since you do test for anal and pharyngeal infection.

'but it can also colonise the ocular, nasopharyngeal, and anal mucosa' has been added. (lines 23 to 24)

Line 65 – consider adding the vaccine name, and at line 72 adding that the OMVs are the same as in MeNZB.

This point is now included (lines 39 and 42)

Line 73- there are now several studies that report potential cross-protection of 4CMenB against Ng that should be referenced.

We have rewritten part of this paragraph and inserted references to Leduc et al (12) and Wang et al (11), where we discuss the protective effect of 4CMenB on Ng infection.

Some details of the recombinant antigens and their surface localisation in Ng would be useful in the introduction as antibody responses to fhbp, GNA1030 and GNA2091 are all described in detail in the results. The absence of NadA in Ng could also be mentioned.

Some more details on the recombinant antigens are given in lines 60 to 64 and the absence of NadA in Ng is now noted.

Materials and Methods

The methodology used is sound, meet the expected standards in the field and there is enough detail provided for the work to be reproduced. The methods are possibly too detailed for the main paper, and it might be good to consider reducing the level of detail if this is available in published papers (e.g., are the details of the cloning, expression, and array printing outlined in Ref 16?) or to have less details in the main paper if a more detailed methods section can be included in the supplementary data.

We have removed the relevant sections describing methods for the extraction and purification of the proteins used for the array and inserted a citation to the Awanye et al. paper which describes these them. This has shortened the Materials and Methods section considerably.

Line 113 – Is data available for past Ng infections in the vaccine cohort?

As noted above, a small minority- 3 of the participants (2 men, 1 woman)- had a laboratory-confirmed Ng infection during the trial- this is now noted in the section on the 4CMenB/Bexsero Trial in Materials and Methods. No other information was available on past histories of Ng infection. However, as noted above, we found that the ‘background’ IgG and IgA reactivities were remarkably stable during the course of the trial. As a result, responses to individual antigens could be readily identified by statistical methods (Tables 1 and 2)- in other words, previous immunological ‘histories’ did not interfere with the analysis of the effect of vaccination on antibody responses.

Line 145 – Consider changing “gonorrhoea was diagnosed” to “Ng infection was diagnosed”.

That has been changed in the manuscript.

Line 148 – Was plasma or serum collected? Typically, serum is analysed in SBA assays, and you later refer to serum.

Both types of samples were collected. Serum was collected from the 4CMenB/Bexsero trial, and plasma was collected from the historical samples. Serum was used for the SBA assay, but this was

not determined for the historical samples. This is now clarified in the manuscript.

Line 188 – check typo for rpm.

Now corrected.

Line 301 – How similar are the main antigens in strain F62 compared to FA1090 used in the arrays?

All antigens in Tables 1 and 2 showed a mean sequence identity of 99.2% between these two Ng strains. F62 was used as a substitute for FA1090 in the SBA analyses because it shows greater sensitivity (see also response to Reviewer 3, below).

Line 330 – Can a reference be added for the collection and processing of PBMCs?

A reference which describes the methods has now been inserted (Landais et al; 33).

Please update ug to µg throughout.

Now corrected.

Results

Line 377-382 and beyond – Can some of these details could be moved to the methods section. It is hard to appreciate the key results in this section due to the complexity of the methods used. Perhaps a clear result and what that means in simple terms for the immune response measure can be stated followed by how it was reached.

We appreciate the challenges inherent in conveying the complexity of multivariate analysis: one advantage of ANOVA Simultaneous Component Analysis (ASCA) is that the results can be presented graphically, separating individual serum samples according to their respective immunoprofiles. We have inserted a fuller description of the method into the relevant methods section, as suggested, and rewritten some parts of the paragraph to which the reviewer refers, to improve intelligibility and clarify the conclusions drawn.

Line 385 – What are the points with the ellipsoids? Are these for the average of all participant responses to all antigens? Can this be clearly explained?

An extension of the ASCA method has been proposed by Liland et al. (reference 38) which introduces confidence ellipsoids to give a guide to relative distributions of points for each independent categorical variable (for the analyses in this manuscript, vaccination stage or disease state). There will therefore be one ellipsoid for each category- t0, t10 or t24 in the case of Fig 1- which is plotted at three different effect level means (40, 68 and 95%). Each ellipsoid has a central point which marks the centre of the distribution of the points for that group in PCA space. The separation of ellipsoids at a given effect level gives an indication of the uncertainty in separation of the categories in the statistical model. For example, the ellipsoid separation is smaller in Fig 1B- where recombinant antigens are removed from the calculation- than in Fig 1A. We consistently applied these effect levels in Figs 1, 2, 7 and 8- this therefore enables the reader to compare effect sizes between IgG and IgA, or the effect of inclusion of the recombinant antigens.

We have changed the relevant part of the text in the Results section to explain this concept more clearly and help guide the reader to an informed interpretation of the results (lines 311 to 347). We have also replotted the lefthand panels in the ASCA plots in all four figures, so that the centres of the ellipsoids are more clearly differentiated from the individual points for each serum sample and made modifications to the figure legends to include reference to the ellipsoid centres.

Line 421 – The meaning of the results regarding Opa are unclear. What is the sequence identity of Opa in the vaccine compared to the Ng sequences?

The value and importance of using Opa and PorB variants on the array is not clear. Vaccine-induced immune responses would presumably focus on the variant in the vaccine, while cross-reactivity to different Ng variants may imply ability to recognise/cross protect against different Ng strains. Are specific variants more common in Ng strains?

10 Opa proteins were identified from our screen of the Ng FA1090 proteome and included in the microarray- they are therefore all from the same strain. The sequence diversity of Opa proteins generally in *Neisseria* has been well documented- they are a family of transmembrane beta barrel outer membrane proteins with two major hypervariable loops. Given their importance in adhesion and, possibly, in infection we thought it important to report on the results. A BLAST search of the genome of the NZ variant (PubMLST ID 34542) identified an allele with sequence identities to the 10 FA1090 Opas which ranged from 60 to 68%. Inspection of our data from the vaccination trial revealed that IgG and IgA responses to individual Opa proteins correlated- examples are now inserted into Supplementary Materials (Fig S4). This observation led us to investigate whether these correlations are reflected in clustering in the tSNE plots in Fig 3B and C: this is indeed the case (Fig S2). An obvious question arising is whether the clustering is attributable to sequence similarity, which would provide further evidence for a subset of cross-reactive IgG or IgA antibodies (Fig S3). This seems to be the case for the OpaD/Opa9/Opa58 grouping and we identified a conserved region in the second hypervariable loop region which could be the target for some cross-reactive antibodies. However, this does not explain the clustering of other Opas. The data do, nevertheless, provide evidence for a subset of cross-reactive antibodies against this important family of outer membrane proteins but which are not induced by the 4CMenB vaccine - an observation which, to our knowledge, is novel. We agree with the reviewer that this implies that such antibodies may cross-react with Opa proteins in other Ng strains, although this remains to be investigated. We have rewritten the text in lines 368 to 389 and inserted an additional figure into Supplementary Material (Fig S4) to make this point more clearly.

There is generally only one PorB allele per Ng strain and it is a major constituent of the outer membrane. Given the central importance of this porin protein (Jones et al Trends Microbiol 2023 (42)), we included 8 PorBs selected for representative sequence diversity on the array. Although these are from different Ng strains, we also observe clustering of antibody responses to PorB (Fig 3B and C). The interpretation of this observation is that, similar to the situation with the Opas, there is a subset of antibodies which cross-react with diverse PorB sequences but, again, these are not induced by vaccination with 4CMenB.

Line 431- Could the reactivity to different Opa variants be due to the conserved regions? If not, does this imply immune reactivity due to Ng infection rather than 4CMenB vaccination?

This is dealt with from the response to the previous comment- the evidence suggests that these antibodies are more likely to be the result of previous exposure, as the reviewer suggests. Given that the vaccinated group is taken from a population with high Ng exposure, this is not surprising.

Table 2 – Stage – “paired t0 with t10t10 or t0 with t24t24” – what does the second t10 refer to in t10t10?

This was a typographical error which has been corrected in the manuscript.

Figure 4 – it would be useful to add IgG and IgA above the graphs on the left and right, or A and B to the panels.

Figure 4 has been modified to separate IgG and IgA into panels A and B.

Line 596 – I suggest changing “patients” to “individuals with a history of Ng infection”, “infected individuals” or similar. Consistency of the terms used should be check throughout the manuscript as this differs in Figures and text.

We have removed reference to the term ‘patient’ and replaced it with more appropriate terms throughout the manuscript.

Line 619 – The phrase “disease-associated individuals” should be changed to something to refer to Ng infection as this is what was measured/included, regardless of disease presentation. Also, for Figs 7 and 8.

These changes have now been made.

Line 621 – “are more important” – it is not clear what this refers to.

Changed to ‘play more important roles’.

Figures 1, 2, 7 – It is hard to appreciate / compare the spread of immune responses as the y-axes are different.

The axes on these three figures, and Fig 8, have no units and are not comparable between each analysis. Thus, in Fig 1A, the y-axis (component 2) only captures about 10% of the variation in the IgG dataset, whereas for the IgA dataset in Fig 1B it is 44%. We set each plot to capture all the data points, so that effectively defines the limits of the plot. We understand the difficulties in comparing different plots- for this reason, we think the ellipsoids are helpful because, as stated above, they give an indication of the uncertainty in the separation of the categories in the statistical model. An improved explanation of the meaning of the ellipsoids is now incorporated into the Results text (as detailed above). In addition, and in response to a point made by Reviewer 2 (below), we have inserted additional descriptions of ASCA and PCA into the second paragraph in the Results section, which should help the reader to interpret these figures.

Discussion

Line 674 – In your previous study, can you please outline what OMV vaccine was used, if this was in humans or animals, and how the findings of that study differed to this one.

The previous study was a Nm, rather than Ng, antigen array applied to measure the IgG responses in a clinical trial induced by an OMV vaccine generated from a genetically modified Nm strain H44/76, with a constitutive expression of the iron transporter FetA. This is now clarified in the manuscript and the reference to the clinical trial inserted (Marsay et al.; 55).

Line 729 – This sentence is unclear. Did you investigate protection against Ng infection in your study? If so, what was the control group and the level of protection seen?

No, protection against Ng infection was not investigated in the study; the sentence has been edited to remove confusion and clarify this point.

Reviewer #2 (Remarks to the Author):

In this manuscript, Stejskal et al. characterized the IgG and IgA antibody responses against *Neisseria gonorrhoeae* protein antigens in a high-risk population of people vaccinated with the 4CMenB anti-meningococcal vaccine. The authors furthermore evaluated the humoral immune responses in a patient population that had been infected with *N. gonorrhoeae*. The authors conclude that the outer membrane vesicle component of the 4CMenB vaccine elicited more durable responses than the recombinant proteins in the vaccine preparation. The authors also speculated that some of the same antigens are also immunogenic during natural infections.

1. In terms of presentation, some of the figures and table would be easier to read/interpret with more information. In Tables 1 and 2, it was not immediately clear what the horizontal line in the table represented. This could be clarified in the footnotes.

The changes to the Results text describing the ASCA analysis, as detailed above for Reviewer 1, should also address this point. The horizontal line separated the t0:t10 and t0:t24 comparisons- this has now been removed to avoid confusion.

Also, why is t10 or t24 duplicated in the last column?

This was a typographical error which has been corrected.

The axes in the figures 1-3 are either not labelled or not clear what they represent.

As detailed above, the results text has been modified to explain in more detail the meaning of the axes in the ASCA plots- the scale is dimensionless and labelled with the appropriate component eigenvector (comp 1 or comp 2). This is standard practice for Principal Component Analysis (PCA). The statement 'Comp1 and Comp2 represent the first and second principal components in the ASCA analysis' has been added to legends of Figs 1, 2, 7 and 8.

The plots in Fig 3 use a different multidimensional reduction method, t-distributed Stochastic Neighbour Embedding (tSNE): although this works in a different way to PCA, the scales on the axes are also dimensionless and particular to each dataset analysed. An explanatory sentence has been introduced into the relevant part of the Results text.

It would help to clarify what the violin plots represent, specifically what is the horizontal spread meant to convey?

In a violin plot, the horizontal spread is indicative of the density of measurements; following the journal's recommendation, we combined violin with box and whisker plots to provide the most complete information on the distribution of data points.

2. Line 49-50: should be "sexually transmitted infection"

This is now corrected; we have also changed the use of the term 'disease' to 'infection' throughout the manuscript, where appropriate.

3. Line 50: Not clear why the authors are explicit about "infection of epithelial cells of the genitourinary tract" specifically when other sites of infection are clearly relevant, even to these studies.

This point was raised by reviewer 1 and has been corrected.

4. Line 51: These statistics are pretty old now, coming from 2016. Much newer data are available.

A more recent reference has been added and the text updated with more recent statistics.

5. Lines 54-55: Not clear what is meant by "the high incidence of multidrug antibiotic resistance has increased concern about the prevalence of Ng infection".

This part of the sentence has been removed and the text rewritten, with the citation of a more appropriate reference to antibiotic resistance in Ng.

6. Line 56: Why are gonorrhoea impacts on the reproductive health of women particularly a concern in Africa and not everywhere?

It is indeed a global concern but the highest incidence of STIs are in Africa- we cite a study which provides evidence for this (Zheng et al, 2022; 3).

7. Lines 86-97: the word protective/protection is duplicative.

That has been addressed and changed in the document.

8. Line 124 and elsewhere: is "enrolment" spelled correctly?

Yes; we have used British English spelling as default in the manuscript.

9. Lines 263-264: It is not clear why EBNA-1 viral protein is a positive control.

In our previous study (Awanye et al) we included EBNA-1 in the microarray because it had been used by other investigators in similar studies as a positive control. However, although it was included in our array, we found that it was not useful as a control in this investigation and in the Awanye study. We therefore omitted the EBNA-1 data from any analysis. We accept that it is confusing to include mention of it when it was not used and have therefore removed it from the manuscript.

10. Did the investigators only use a single dilution (1:300) of human serum? This could limit the analysis severely as some antigens may not be detected at this dilution.

From previous experience in the laboratory, although the dilution of the sera could range from 1:50 up to 1:1000, 1:300 has shown the best balance between foreground and background signal. Hence it was empirically chosen as the working dilution. It is true that there could be some antigens that were not detected at this dilution but if we had used more dilute sera we would have introduced more background noise in the signal (saturation of the nitrocellulose with a higher number of antibodies bound to the higher responding antigens).

11. A significant concern for the entire analysis is the level of sequence variation between the N. meningitidis vaccine antigens and the FA1090 or F62 gonococcal strains used for testing. It is not clear to what degree sequence diversity between antigens can be distinguished from degree of “antigenicity”. Throughout the presentation, the authors refer to “the magnitudes of responses to each individual antigen” (line 649) but the amount of signal is related to both the magnitude of the response and the degree of sequence conservation. Little to no attention was given to the latter.

The investigation is based on the observation, now established from several studies, that vaccination with 4CMenB/Bexsero has a protective effect on Ng infection. A central aim of the study is therefore to examine the cross-reactivity between antibodies generated by 4CMenB and Ng antigens. We have recorded the levels of sequence identity between antigens in 4CMenB and their Ng counterparts in strain FA1090 in Tables 1 and 2. The reviewer is correct that, on the basis of antigen reactivity profiling alone, it is not possible to distinguish intrinsic ‘antigenicity’- or propensity to induce an immunogenic response- from sequence variation between any given antigen in 4CMenB and its Ng counterparts in different Ng strains (including FA1090). However, the basis of the study is a series of statistical analyses which use these antibody reactivities to distinguish between different groups (eg pre- and post-vaccination) and identify the antigens which are responsible for driving those differences. The reasons underpinning such differences will be a combination of factors, including sequence diversity, intrinsic immunogenicity and, indeed, other factors, such as expression levels. The relevance of the study lies in the identification of the most important Ng antigens in a population where the incidence of gonorrhoea is high.

12. In several instances (also highlighted as – in % identity tables 1 and 2) the authors indicate that antibody responses were detected against a gonococcal antigen even though no meningococcal sequence homolog was detected. It is not clear how these responses then would be generated. These observations raise some doubt about the analysis unless the authors have a reasonable explanation, which was not provided.

The reviewer has raised an important point. Tables 1 and 2 list antigens against which cross-reactive IgG and IgA antibodies are induced by vaccination with 4CMenB. To exhibit cross-reactivity, an antigen in 4CMenB induces a subset of B-cells to produce antibodies which recognise epitopes in cognate Ng antigens within the microarray. However, the degree of sequence identity is only an approximate guide to the likelihood of cross-reactivity. It is a value which applies to the entire sequence but does not take account of the possibility for localised regions of sequence and/or structural similarity which form conserved epitopes. An example where this may be the case is the final entry in Table 2- a membrane protein which contains an OmpA-like domain (NEIS2704). We have inserted a brief acknowledgement of this issue into the text:

‘Several antigens in Tables 1 and 2 did not have a close meningococcal homolog, raising the question why IgG or IgA antibodies were generated against them. One explanation is that this

cross-reactivity may be caused by for localised regions of sequence and/or structural similarity which form conserved epitopes, possibly for functional reasons. Examples of antigens where this might be the case are membrane protein 1 (NEIS2704) which contains an OmpA-like domain, or NGO1847 (NEIS2724) which contains a Tetratricopeptide Repeat (TPR) repeat. TPR is a structural motif which is widespread in bacterial pathogens' (a reference to Cerveny et al., which reviews this topic, is inserted (43)).

13. Line 469: This observation must correspond to Figure 3 (not Figure 2) as Figure 2 has no panel C.

Now corrected in the manuscript.

14. Line 513: What is meant by "intrinsic and extrinsic membrane proteins" that are transmembrane beta-barrel structures?

The sentence has been rewritten to avoid confusion.

15. Lines 514-516: What is meant by "the inclusion Indicates the importance of outer membrane protein folding and insertion"?

The sentence has been rewritten to clarify the point.

16. Speaking of membrane folding, how can it be ensured that the purified proteins in the microarray are correctly folded? If this cannot be assured, then it is likely that many antibodies against conformational epitopes have been missed in the analysis.

The reviewer is referring to the subset of antigens which are integral outer membrane proteins (iOMPs). Most of these proteins have been worked on by the Derrick group for many years, notably Opc, the Opas, PorB and TonB-coupled transporters such as FetA. In these cases, we have published structural and/or functional data to support their correct folding. Many iOMPs are listed in Tables 1 and 2- evidence that sufficient epitopes are preserved to satisfy the statistical screen for antigens stimulated by vaccination with 4CMenB. In addition, preservation of all conformational epitopes is not necessary for the ASCA and tSNE analyses presented. These use data from all antigens in the microarray, but it is not required that such datasets are comprehensive, merely that they are sufficiently complete to allow systematic differences between groups (eg pre- and post-vaccinated) to be identified.

17. Line 520: Because pilE is subject to high-frequency antigenic variation, the sequence identity (listed as 66%) is not fixed when comparing the 4CMenB strain and FA1090.

The sequence identity was calculated using the sequence database entry for the 4CMenB strain and the actual expressed sequence for the PilE variant used in the Ng antigen microarray. Sequence identities were included to help provide the reader with a guide to the degree of sequence similarity between the Nm antigens in the 4CMenB strain and their Ng homologs in FA1090. PilE is one of the more variable antigens listed in Tables 1 and 2; in general, there does not seem to be a strong correlation between stimulation of cross-reactive antigens by 4CMenB and

their sequence conservation between Nm and Ng. This observation suggests cross-reactivity against conserved epitopes- we explored this more fully with the Opa and PorB variants.

18. Given the technical issues with the ELISpot analysis, the data seem premature for this presentation.

T-cell responses are likely to play an important part in a protective response against Ng infection, although current understanding of them is poor. ELISpot assays are often used but there are challenges in adapting them to examine iOMP antigen-specific stimulation, given that integral membrane proteins often require solubilization by detergent, which is not readily compatible with the assay. One way forward is to use T-cell epitope prediction to identify peptides but, as we show here, this was unsuccessful. The value of inclusion of this information, which only takes up a small part of the manuscript, is that it is potentially useful to other investigators who will likely have to devise more sophisticated means to solve this problem.

19. Line 616: HpuB does not seem to be represented in Figure 7, unlike what is indicated in the text.

Thanks to the reviewer for spotting this error- we were referring to transferrin binding protein A, which features in the IgG and IgA analyses. This is corrected and a suitable reference cited.

20. With respect to the natural infection study, how can the impact of multiple previous infections be taken into account in terms of antibody “levels” detected? Also, this analysis is even more problematic with respect to the divergence of infecting strains, their sequence conservation with each other and with the type strain used here (FA1090).

Subjects in the natural infection study are drawn from the same population as the vaccine trial, but the serum samples were collected over a period of 10 years, during which they would have been exposed, presumably, to different Ng strains, probably on multiple occasions. Our data show that individual subjects have significant antibodies against Ng antigens even before infection- this corresponds to the pre-vaccination group t0 and the ‘Before’ group in the natural infection study. These background immunoprofiles are remarkably stable and we essentially subtract them out when comparing them against t10/t24 or During/After, thus identifying antigens against which IgG or IgA responses were stimulated by vaccination or infection.

FA1090 was used as the reference strain for antigen production because it is well-characterised and used in the field. This will allow us and other investigators to make comparisons with human and animal infection models in the future. For example, the same strain is being used in a human challenge trial with 4CMenB (ClinicalTrials.gov Identifier: NCT05294588). Clearly, any single strain will diverge to a greater or lesser extent from the diversity of strains circulating in the Kenyan population. However, the basis of our analysis is to use the array to record the relative increase in responses to specific antigens following vaccination or disease. The choice of reference strain is less important because we are not ascribing significance to absolute antibody levels.

Reviewer #3 (Remarks to the Author):

The study Rationale and objectives are to use the meningococcal B vaccine (4CMen B/Bexsero) as an immunological probe to identify *N. gonorrhoeae* (Ng) antigens by cross-reactive responses in individuals with high Ng exposure as recurrent exposure to Ng may elicit some immunologic memory to support Ng vaccine development. The primary objective is to assess if immunisation of individuals at risk for gonococcal infection with 4CMenB (Bexsero) elicits humoral and T cell cross-reactive responses against Ng.

The design is an open label, single arm study of 2 doses of Bexsero, which contains the MeNZB OMV component plus three recombinant antigens (NadA, fHBPgNA2091, and NHBA-GNA1030); there is a high level of sequence identity exists between MeNZB OMV (the vaccine and Bexsero OMV antigens, and gonococcal proteins. NHBA is the only 4CMenB vaccine (Bexsero®) recombinant antigen that is conserved and surfaced exposed in Ng.

The study population is a cohort of MSM and FSWs in coastal Kenya which has a high incidence of GC infection: incidence estimates for Ng (based on Xpert testing) in HIV-negative 18-25 year-old MSM participants is 21.4 per 100 person-years (95% CI: 13.1-35.0), and 43.5 per 100 person-years (95% CI: 23.4-80.8) in HIV-infected individuals

Immune response methods: Comparison of the pre/post immunization sera will provide a comprehensive overview of Bexsero antigens which induce Ng-cross reactive IgG responses and could therefore be responsible for Nm OMV-mediated protection against Ng. Samples with cross-reactive antibodies against the OMV antigens, which may offer protection against Ng will be selected to inform antigen choice and vaccine design.

Humoral responses: serum antigen- and OMV-specific IgG1, IgG2a, Ig3A and IgA titres will be determined by ELISA approximately two weeks after final immunizations. ELISA assay using peptides will covering recombinant protein antigens, purified antigens, and defined OMVs Responses will be compared among pre-post-test groups by repeated measures.

Th1 responses: all candidate vaccines that show efficacy in the mouse models against Ng (2C7 peptide mimetic, Gc OMVs administered with IL-12) induce Th1 responses.

Comments for the authors:

Abstract: 1) "add to protect against infection and reproductive health consequences" to "driven by antimicrobial resistance, 2) add 'single arm open label study of 2 doses of Bexsero and humoral and cellular immune response was measured at 4 time points over six months in which pre-post immunization time points were compared by a T test". 3) Clarify "serum IgG and IgA reactivities against the gonococcal homologs of the recombinant antigens in the vaccine peaked at 10 weeks but declined by 24 weeks. The reverse was the case for most antigens originating from the OMV component" and why that leads to their conclusion that cross-protection of the 4CMenB vaccine against gonorrhoea could be explained by cross-reaction against a diverse selection of antigens derived from the OMV component."

We have incorporated the changes suggested by the reviewer into the Abstract.

In response to point 3, there is evidence, established by other investigators, that 4CMenB/Bexsero provides protection against Ng infection. It is reasonable to propose that the antibody responses to antigens in Bexsero could play a role in providing protection. Our study breaks down IgG and IgA responses to specific Ng antigens following vaccination with Bexsero; in particular, we note that responses to the recombinant protein antigens (eg Factor H binding protein) peak earlier and are more short-lived than those against the OMV antigens. Given that protection against Ng infection

from Bexsero has been observed to last well beyond the shorter timepoint in our study (10 weeks), it is reasonable to infer that a combination of antigens within the OMV component could contribute to the protective response.

Introduction:

1) Lines 64-66: would clarify that this was not a randomized evaluation of Bexsero and rather a retrospective observational cohort finding.

That has been addressed and changed in the manuscript (line 38).

Methods:

1) Lines 146-148: In the historical cohort, there were 85 NG infections. Please indicate how many of the 50 participants had a prevalent Ng infection at baseline and how many were negative at baseline and had 1 or more incident Ng infections?

The 50 participants were from the Bexsero study, not the historical cohort; none of the participants in the Bexsero trial had a Ng infection at baseline, so all were negative. Only three had a history of previous (i.e. laboratory confirmed) Ng infection.

2) Lines 161: How closely related is the gonococcal strain FA1090 to circulating Ng strains in Kenya?

This is an interesting question but the data available to answer it are limited. Previous analysis of Ng strains in Kenya categorised isolates into three clusters (Cehovin et al JID 218,801- this is cited as reference 21 in the revised manuscript). FA0190 was selected as the strain for generation of the microarray because it is well characterised in the literature. However, we would argue it is very unlikely that genetic variation among gonococci will materially affect our conclusions. In the case of the Bexsero study, we are looking at the induction of cross-reactive antibodies between Nm and Ng- multiple antigens fulfil the stringent criterion for inclusion in Tables 1 and 2, demonstrating that many epitopes are sufficiently well conserved to exhibit cross-reactivity, in spite of sequence differences. In addition, we observe high levels of 'background' reactivities in serum samples (eg t0 in Fig S1), demonstrating that this population has variable but significant IgG and IgA antibodies against the Ng FA1090 antigen variants on the array. This conclusion is also borne out by the results we obtained with PorB variants from 8 diverse Ng strains- again, a high level of cross-reaction is exhibited in clustering within tSNE plots (Fig 3). Overall, our data point to the general conclusion that our results are not likely to be greatly affected by choice of 'parent' strain from which to derive antigen sequences for the array.

3) Lines 277-292: Were paired plasma samples analyzed on the same microarray slides to minimize variability in immune response due the amount of protein on different slides?

Stringent quality control processes were employed in fabrication of the array slides by a contractor (Arrayjet) which ensured that every array in each slide contains the same amount of protein printed. In addition, we used serial dilutions of a human antibody as a standard in each array, to ensure consistency of detection and linearity between different slides. In practice, we found array to array and slide to slide variation was minimal. We also conducted batch controls where

randomised samples were tested in different slides with the same conditions, and there were no significant changes in the datasets.

4) Line 302: Why was Ng strain F62 selected for the SBA? Could a panel of Ng strains been evaluated for SBA?

Ng strain F62 exhibits better sensitivity in the SBA assay than FA1090 and is routinely used when testing 4CMenB/Bexsero in a mouse infection model. As noted above, the antigens identified in Tables 1 and 2 had over 99% sequence identity between the two strains. In principle, a panel of strains could have been analysed but, given the fact that we failed to identify a relationship between IgG or IgA responses to individual antigens and bactericidal titres, we think it would have been unlikely to be informative.

5) Justify the 6 months follow-up and whether that is sufficient to assess chronology and durability of different immune responses.

The t24 timepoint was certainly sufficient to show the difference in antibody responses between the recombinant and OMV-derived antigens (Figs 4, 5, 6). We note that other investigators have presented evidence that the protective effect of Bexsero is sustained for 2-3 years.

6) Given the potential that HIV infection could confound the immune response, justify including both persons living with HIV and HIV uninfected: Should the results be analyzed separately?

It should be noted that participants living with HIV were on antiretroviral therapy, and their viral load was suppressed. We found little evidence to suggest that HIV infection affected the IgG or IgA immunoprofiles. In response to the reviewer's comments, Principal component analysis (PCA) was carried out on the t10-t0 and t24-t0 datasets, which quantify the stimulation of IgG or IgA responses at the 10 and 24 week timepoints respectively. We can use these datasets to separate the immunoprofiles for each vaccine recipient: plots for the first two principal components are now shown as new figures in Figs S5 & S6. Each point is derived from the t10-t0 and t24-t0 immunoprofiles for each vaccinee; points are coloured by either HIV status (positive/negative) or sex (male/female). Ellipses are superimposed at 95% probability. There is little evidence for separation between the groups, for IgG or IgA, at either timepoint, with the possible exception of HIV status at t10-t0, for IgG (Fig S5, top left). However, even in this case, there is considerable overlap between the HIV positive and negative groups. We would not discount the possibility that HIV infection could affect responses to Bexsero but this would require further study.

We have inserted a short paragraph to this effect in the main Results section, which describes the PCA analysis.

7) Given that this is a highly exposed cohort, did they stratify by whether they had NG immune response at baseline?

The antigen-specific responses are recorded at baseline but are then subtracted from future timepoints to identify, by different methods, which groups of antigens are targeted following vaccination or disease.

8) Did they do a sensitivity analysis in which they excluded those who acquired CT during follow-up

as that could have modified their immune response to Ng (based on observational data from New Zealand)?

The reviewer raises an interesting point- whether immune responses might be affected by acquisition of CT infection following vaccination with Bexsero. Only two participants acquired CT infections after immunization, a number too small to conduct any meaningful statistical analysis. We did examine whether the immunoprofiles of these two individuals were unusual, through analysis of PCA and tSNE plots- this was not the case.

9) Did the investigators adjust for multiple comparisons given the large number of antigens assessed?

Yes, a modified Bonferroni method, after Hommel (1988) was used; it is cited in the footnotes to Tables 1 and 2 and in the relevant section in Materials and Methods.

Results:

10) Lines 367-8: Clarify whether there was only a single Ng infection during follow-up (e.g., any infections earlier than the one described at month 6)?

Yes, only 1 Ng infection was identified in follow-up.

11) Lines 546-557: Did you assess the viability of the PBMCs, given that collection, storage and shipping could have reduced their viability?

PBMCs had a mean viability of 97%- a sentence to this effect is now inserted in the relevant section of Materials and Methods.

Discussion:

13) Lines 406-418 and Fig 3 ABC: Discuss whether the different IgG and IgA reactivities, the higher IgG than IgA reactivity at t24, and the highest reactivity to NHBA and GNA2091 was also seen in your study of Bexsero in the lower risk cohort in the UK. This would be of interest in terms of whether this is a consistent finding in a cohort with lower risk of previous exposure to Ng.

The data in Fig 3A compares the profiles of IgG and IgA reactivities against the antigen panel: rather than comparing absolute values, the data show that IgG or IgA responses to particular antigens are overlapping but not identical. We have not conducted any other studies of Bexsero-vaccinated individuals with a Ng antigen microarray although it would certainly be valuable, as the reviewer suggests, to compare the results in a population with lower exposure.

14) Lines 541-2: What are the implications of lack of evidence of IgA or IgG responses to individual antigens and SBA?

We thought it important to note that we did not observe any correlations between bactericidal activity and any IgG or IgA responses. SBAs are well established as a correlate of protection against Nm infection but the situation with Ng is less clear. The value of such an approach is that it offers a way to identify particular antigens which are associated with bactericidal activity and therefore potential vaccine components. We have shown that log(SBA) can be used to train a statistical model using IgG reactivities against a Nm microarray panel (Chang et al (18)) but the data presented here suggest that this is unlikely to be successful, at least with this dataset.

15) Line 648: Clarify that you are comparing the Bexsero responses to a similar Kenyan population who had Ng infection and samples obtained during and after acquired infection.

This has been addressed and changed in the manuscript.

16) Lines 743-745: Can you propose criteria for prioritizing the different antigens in Tables 1 and 2: strength of response, durability at T10 and T24, and/or correlation between IgG responses to different antigens which have been observed in this and prior studies of humans immunized with Bexsero? What are the next steps?

Prioritization of antigens for inclusion in a Ng vaccine is dependent on several considerations. This study has informed the process but there are still important, fundamental questions to be resolved. For example, as the reviewer mentions, the observation of differential responses in the OMV-derived antigens, compared with the recombinant proteins, at the t10 and t24 timepoints requires further investigation. It indicates that the context within which an antigen is presented- eg an OMV environment- plays a role in the subsequent immune response against it. Consideration also needs to be given to the diversity of sequences within Ng surface antigens. More information is also needed on antibody responses to 4CMenB/Bexsero, using Nm and Ng antigen arrays, in different populations- the reviewers have raised the important question of prior immunity and exposure to previous Ng infections. Again, more extensive work will be needed to evaluate how genetic (eg haplotype) and other factors may interact to influence immunogenic responses to Bexsero and other candidate Ng vaccines.

REVIEWERS' COMMENTS

Reviewer #2 (Remarks to the Author):

The revised manuscript appears to address the reviewers' comments adequately, thus resulting in a much improved presentation.